# Group 3 medulloblastoma transcriptional networks collapse under domain specific EP300/CBP inhibition

Noha A. M. Shendy[1,11], Melissa Bikowitz[2,3,11], Logan H. Sigua[4,11], Yang Zhang[5], Audrey Mercier[6], Yousef Khashana[1], Stephanie Nance[1], Qi Liu[4], Ian M. Delahunty[1], Sarah Robinson[6], Vanshita Goel[6], Matthew G. Rees [7], Melissa A. Ronan [7], Tingjian Wang[4], Mustafa Kocak[7], Jennifer A. Roth [7], Yingzhe Wang [8], Burgess B. Freeman [8], Brent A. Orr [9], Brian J. Abraham [5], Martine F. Roussel [6], Ernst Schonbrunn [2,3] ✉, Jun Qi [4,10] ✉ & Adam D. Durbin [1] ✉

Chemical discovery efforts commonly target individual protein domains. Many proteins, including the EP300/CBP histone acetyltransferases (HATs), contain several targetable domains. EP300/CBP are critical gene-regulatory targets in cancer, with existing high potency inhibitors of either the catalytic HAT domain or protein-binding bromodomain (BRD). A domain-specific inhibitory approach to multidomain-containing proteins may identify exceptional-responding tumor types, thereby expanding a therapeutic index. Here, we discover that targeting EP300/CBP using the domain-specific inhibitors, A485 (HAT) or CCS1477 (BRD) have different effects in select tumor types. Group 3 medulloblastoma (G3MB) cells are especially sensitive to BRD, compared with HAT inhibition. Structurally, these effects are mediated by the difluorophenyl group in the catalytic core of CCS1477. Mechanistically, bromodomain inhibition causes rapid disruption of genetic dependency networks that are required for G3MB growth. These studies provide a domain-specific structural foundation for drug discovery efforts targeting EP300/CBP and identify a selective role for the EP300/CBP bromodomain in maintaining genetic dependency networks in G3MB.

Cellular gene expression programs are controlled in part by the coordinated function of epigenetic enzymes[1–3]. These enzymes establish chromatin marks that are associated with active promoter and enhancer elements, and which control downstream gene expression by facilitating or repressing the activity of RNA polymerase and associated complexes[1–3]. Two key proteins that function to maintain and reinforce malignant gene expression programs through regulation of transcription are the multidomain and paralogous histone

[1]Division of Molecular Oncology, Department of Oncology, St. Jude Children's Research Hospital, Memphis, TN, USA. [2]Drug Discovery Department, Moffitt Cancer Center, Tampa, FL, USA. [3]Department of Molecular Medicine, Morsani College of Medicine, University of South Florida, Tampa, FL, USA. [4]Department of Cancer Biology, Dana-Farber Cancer Institute, Boston, MA, USA. [5]Department of Computational Biology, St. Jude Children's Research Hospital, Memphis, TN, USA. [6]Tumor Cell Biology Department, St. Jude Children's Research Hospital, Memphis, TN, USA. [7]The Broad Institute of MIT and Harvard, Cambridge, MA, USA. [8]Preclinical Pharmacokinetics Shared Resource, St Jude Children's Research Hospital, Memphis, TN, USA. [9]Department of Pathology, St Jude Children's Research Hospital, Memphis, TN, USA. [10]Department of Medicine, Harvard Medical School, Boston, MA, USA. [11]These authors contributed equally: Noha A. M. Shendy, Melissa Bikowitz, Logan H. Sigua. ✉e-mail: Ernst.Schonbrunn@moffitt.org; jun_qi@dfci.harvard.edu; adam.durbin@stjude.org

acetyltransferases (HATs), EP300 and CBP[4–7]. EP300 and CBP broadly regulate the activity of other proteins through their protein acetyltransferase catalytic activity[8–10]. These proteins contain several highly homologous domains, including KIX, bromodomains (BRDs) and HAT domains, through which they interact with, dock to, and acetylate target proteins, respectively[5,10,11]. Evidence from knockout studies in the mouse[12–15], and studies in cancer cells[5,16–18] have implicated these proteins as critical to the development of normal tissues and indeed, as potential targets for therapeutic development in disease states. As a result, significant efforts by many have led to a host of small molecule inhibitors and degraders useful to interrogate disease biology driven by EP300/CBP[5,10,19–27].

One function of EP300 and CBP is to coordinate gene transcription[28–30]. EP300 and CBP proteins bind acetylated histone H3 at enhancer sites through their BRDs, and establish separate histone acetylation marks using their catalytic HAT domains[10]. Both of these activities are required for efficient enhancer activity[31]. To this end, combinations of HAT and BRD probe treatment has synergistic effects on disrupting gene expression[32], suggesting that each domain of EP300/CBP may contribute to different aspects of transcriptional control. Due to the homology between these subdomains in both EP300 and CBP, small molecule inhibitors often display expected on-target toxicities related to inhibition of both protein species. This implies that a broad-based search for tumor types with exceptional responses to domain-specific EP300/CBP-targeted therapeutics would be a high-yield approach to identifying tumors for specific inhibition. Given observed on-target toxicities associated with inhibition of both EP300 and CBP, therefore, this approach would be predicted to maximize a potential therapeutic window for clinical translation[5,22].

Here, we identify and separate the effects of targeting distinct domains of EP300 and CBP proteins across a panel of 460 cancer cell lines representing 31 distinct tumor types. We identify that the high-risk pediatric embryonal brain tumor medulloblastoma (MB) is exceptionally sensitive to EP300/CBP BRD inhibition, compared with HAT domain inhibition. These effects are associated with rapid and selective loss of expression of a dense network of genes required to maintain Group 3 medulloblastoma (G3MB) cell growth, including the medulloblastoma driver oncogene c-MYC. Using crystallography, we identify the binding mechanism of the EP300/CBP-specific BRD inhibitor CCS1477 in complex with the EP300, CBP and BRD4 BRDs, and define components of the CCS1477 catalytic binding core that are required for compound activity. Our findings identify specific tumor contexts to examine domain-specific inhibition of EP300/CBP, and provide structural and molecular bases for EP300/CBP BRD inhibition in high-risk, Group 3 medulloblastoma.

## Results

### Cancer cell lines are broadly sensitive to EP300/CBP inhibitors

Several small molecule inhibitors of distinct protein domains of EP300/CBP are available, though the majority target the histone acetyltransferase domain (HAT) or bromodomain (BRD)[5,10,21,22]. Recently, nearly equipotent inhibitors of each of the HAT and BRD have been developed: the spirooxazolidinedione A485[21,22] and the dimethylisoxazol-benzimidazole CCS1477[5]. These compounds inhibit the HAT or BRD functions of EP300/CBP, respectively, with low nanomolar potency, are cell permeant, and are largely specific for EP300/CBP over related HAT- and BRD-containing proteins[5,21,22]. Thus, these compounds provide a unique opportunity to compare the effects of domain-specific inhibition of EP300 and CBP. We used these tools to investigate the relative contribution of the EP300/CBP BRD or HAT domain to tumor cell growth (Fig. 1a). To do so, we performed 10-point, dose-response growth assays using mixed pools of 460 barcoded cancer cell lines, over five days of treatment in vitro[33,34]. Growth was measured using an area-under-the-curve (AUC) approach (Supplementary Table 1, Supplementary Fig. 1a). Comparison across all

cancer cell lines indicated that the HAT inhibitor A485 had a greater effect on growth suppression than the BRD inhibitor CCS1477 (Supplementary Fig. 1b), and that the individual effects of CCS1477 and A485 across all cell lines were not well correlated ($R^2 = 0.13$). Given potential differences in compound penetration and solubility, and to compare the relative cell line and tumor-specific effects of these two compounds in assays performed non-simultaneously, we performed a median normalization of each dataset (Supplementary Table 2). This had no effect on the correlation of individual compound activity ($R^2$ remained 0.13), but provided more comparable AUC ranges between A485 and CCS1477 treatments (Supplementary Fig. 1c). Next, we sought to use this to examine the relative cell line-specific response to either A485 or CCS1477. To do so, we directly compared each individual cell line by analyzing the median normalized ratio (CCS1477/A485) (methods) on a per-cell-line basis. The median-normalized AUC ratio of CCS1477/A485 demonstrated that the majority of cell lines were nearly equivalently affected by HAT or BRD inhibition, with a fraction of cell lines displaying median normalized ratios of >1.2 (13.7%, 63/460) or <0.8 (10.6%, 49/460) (Supplementary Fig. 1d). To orthogonally validate these findings, we examined the effects of A485 and CCS1477 on the growth of cell lines from distinct tumor types that displayed a range of responses to these two compounds. Low-throughput testing in TE617T and RhJT rhabdomyosarcoma, NCIH650 and NCIH2122 non-small cell lung carcinoma, 143B osteosarcoma and Kelly neuroblastoma cell lines (Fig. 1b–e, Supplementary Fig. 1e,f) demonstrated, as predicted, that NCIH650, RHJT and 143B cells were more sensitive to CCS1477 than A485 (AUC ratio (CCS1477/A485): 0.68 NCIH650, 0.67 143B, 0.76 RHJT) while in contrast, TE617T, NCIH2122 and Kelly cells were more sensitive to A485 than CCS1477 (AUC ratio (CCS1477/A485): 1.13 TE617T, 1.68 NCIH2122, 1.43 Kelly). These data provide support that the results of these screening experiments are reproducible both within individual lineages (non-small cell lung carcinoma and rhabdomyosarcoma) as well as between a variety of lineages.

Next, we examined whether specific characteristics of cell lines predicted enhanced effects of either CCS1477 or A485. To do so, we integrated expression, mutation and dependency data from the Cancer Cell Line Encyclopedia (CCLE) and the Cancer Dependency Map (DepMap) to investigate whether mutational status of EP300 or CBP predicted differential response to CCS1477 or A485. Univariate analysis of gene expression, mutation and exome-wide DepMap CRISPR-cas9 dependency did not reveal clear associations between these variables and the median normalized AUC value. Since prior data had demonstrated that cells with mutations in CBP were more sensitive to loss of EP300[18], we next examined CCLE data for the mutational status of EP300 and CBP. These findings demonstrated that mutational status had no relationship with median-normalized AUC, indicating that the differential response to CCS1477 or A485 was not associated with individual or combined mutational status of the target (Supplementary Fig. 1g). Then, we sought to identify whether there was a correlation between the median normalized AUC ratio and genetic dependency on either EP300 or CBP, using DepMap CRISPR-cas9 knockout data[35,36]. As predicted by the known differences between chemical inhibition of two proteins (both EP300 and CBP), and genetic loss of a single protein species (EP300 or CBP), there was no association between the individual CRISPR-cas9 knockout of EP300 or CBP and the relative effect of BRD or HAT domain inhibition (Supplementary Fig. 1h, i). These data indicated that the relative susceptibility to EP300/CBP BRD vs HAT domain inhibition, as determined by the ratio of median normalized AUC is not determined by a single driver and may be multifactorial in nature.

Next, we sought to identify whether specific tumor types displayed enhanced sensitivity to A485 or CCS1477. Cell lines were collapsed into 31 distinct tumor types, excluding those with $n < 3$ cell lines (Fig. 1f), and sorted by median-normalized AUC ratio. This yielded 454

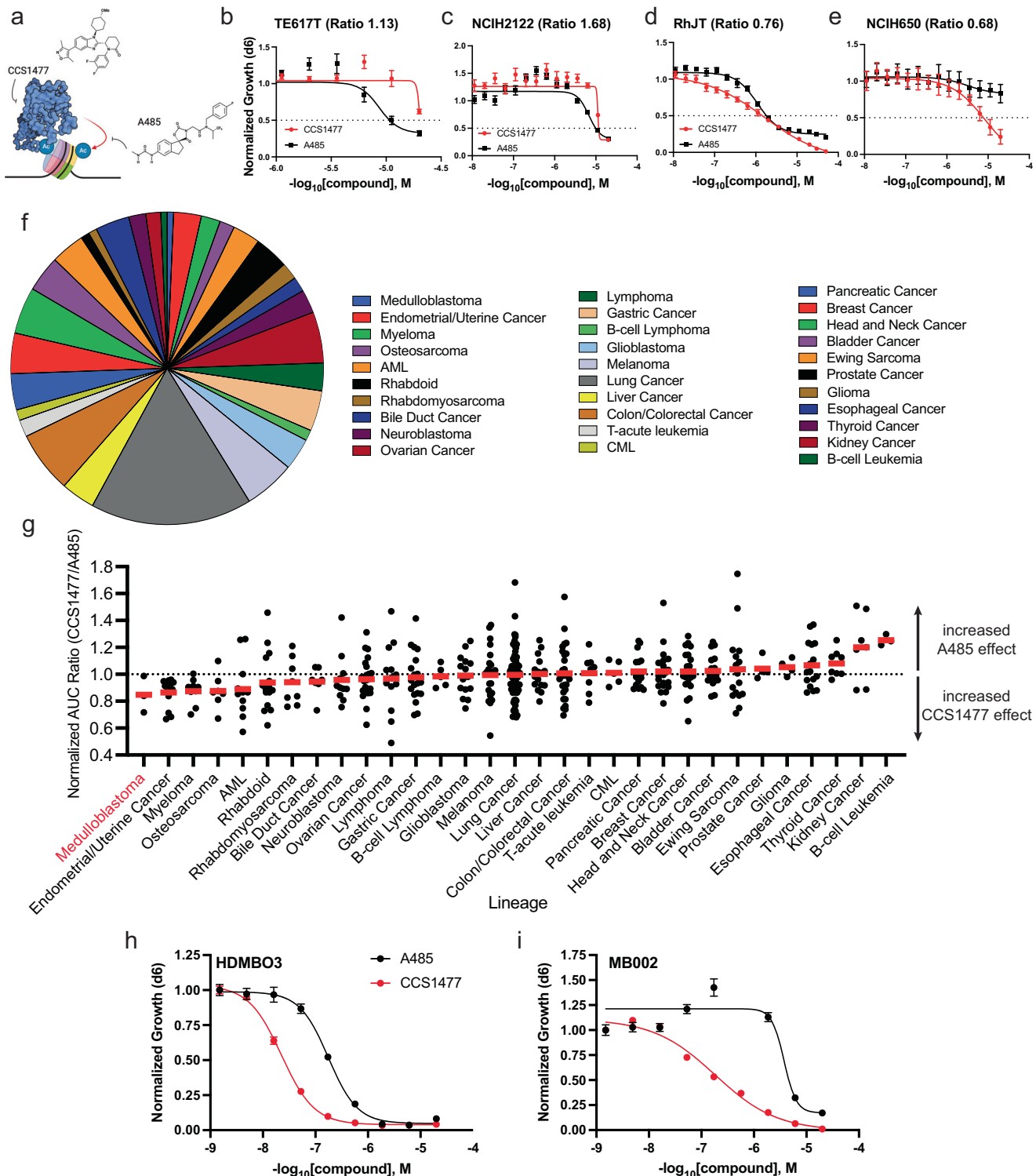

**Fig. 1 | Chemical targeting of the bromodomains of EP300/CBP causes reduced cell growth in medulloblastoma cells, compared with HAT domain targeting.**
**a** Schematic of compound targeting of EP300 and CBP proteins by either CCS1477 (bromodomain) or A485 (HAT domain)-targeted compounds. Figure created in Biorender.com. EP300 catalytic core structure retrieved from PDB: 6K4N (PDB https://doi.org/10.2210/pdb6K4N/pdb). **b**–**e** TE617T (**b**), NCIH2122 (**c**), RhJT (**d**), and NCIH650 (**e**) cells were tested for dose-response effects of CCS1477 and A485 after 6 days by Cell-Titer Glo assay. $n = 3$ independent biological replicates for each dose. Error bars represent S.E.M. Ratio = median normalized AUC ratio. Source data are provided in Source Data file. **f** Distribution of 454 cancer cell lines subjected to

PRISM screening with either CCS1477 or A485. **g** Normalized AUC ratio values for 454 cell lines tested for CCS1477 or A485 effects in dose response after 5 days. Dots reflect individual cell lines. Dotted line indicates normalized AUC value of 1, where normalized effect of CCS1477 is equal to A485. Arrows denote higher or lower AUC values, representing increased relative effects of A485 (higher) or CCS1477 (lower). Red bars indicate median value of lineages. $n = 454$ cell lines, 31 tumor types. Source data are provided in Source Data file. **h, i** HDMBO3 (**h**) and MB002 (**i**) group 3 medulloblastoma cell lines were tested for dose-response effects of CCS1477 and A485 after 6 days by Cell-Titer Glo assay. $n = 3$ independent biological replicates for each dose. Error bars represent S.E.M. Source data are provided in Source Data file.

cell lines for analysis. By this metric, most tumors were similarly inhibited by A485 or CCS1477, with a median normalized AUC ratio of -1 (Fig. 1g). Few tumor types, including medulloblastoma, endometrial/ uterine cancer and multiple myeloma displayed a reduced median normalized AUC ratio, indicating an increased relative effect of CCS1477, compared with A485 (Fig. 1g). These data are consistent with recent published reports demonstrating CCS1477-driven responses in clinical trials of patients with hematologic malignancies, including multiple myeloma[37]. In contrast, others such as B-cell leukemia, kidney cancer and thyroid carcinoma displayed enhanced relative effects of A485, compared with CCS1477 (Fig. 1g). These data indicated that some tumor types may display differential sensitivity to domain-specific inhibition of EP300/CBP.

## Medulloblastoma shows enhanced sensitivity to bromodomain inhibition of EP300/CBP

CCS1477 is an EP300/CBP-specific BRD inhibitor[5], and has entered clinical trials for adult patients with metastatic carcinomas and advanced hematologic malignancies (NCT04068597, NCT03568656)[37]. Examination of tumor-specific responses, however, revealed that CCS1477 demonstrated greatest activity, relative to A485, in cell lines derived from the high-risk pediatric brain tumor, medulloblastoma (Fig. 1g). Medulloblastoma (MB) is an aggressive pediatric malignant brain tumor composed of distinct disease subtypes regulated by different driver oncogenes[38–41]. Classically, MB tumors are characterized by four subgroups, Sonic hedgehog (SHH)-activated, WNT-activated, Group 3 and Group 4[40,42,43]. Two of three tested MB cell lines, ONS76 and UW228 cells were preferentially sensitive to CCS1477, as compared with A485 (Fig. 1g). These cell lines are reported to be *SHH*-activated MB cell lines with high expression of *MYCN*[44,45].

One of the most aggressive MB tumor subtypes are Group 3 medulloblastoma (G3MB), which are commonly characterized by overexpression of the driver oncogene *c-MYC* due to gene amplification in 17% of cases[40,42,43]. These tumors are associated with poor patient survival[38]. Given prior reports linking CCS1477 to disruption of *c-MYC* expression[5], we sought to further explore these findings in higher risk subtypes of MB, and in models that more closely pattern in vivo gene expression states. Thus, we performed a comparative analysis of CCS1477 and A485 effects by CellTiter-Glo analysis using two G3MB cell lines, HDMB03 and MB002, growing in neurosphere cultures[46]. Both HDMB03 and MB002 cells displayed dramatically enhanced sensitivity to CCS1477, as compared with A485 (Fig. 1h, i). Given these findings, we next performed preclinical pharmacokinetic (PK) analysis of CCS1477 after either 25 or 50 mg/kg i.p. dosing. These results demonstrated similar plasma PK properties as previously reported (Supplementary Fig. 2)[5]. Since medulloblastoma is a primary brain tumor, we explored the blood-brain barrier penetration of CCS1477 in our mice. We sacrificed treated mice at three timepoints after dosing (8, 16, 24 h) and perfused the murine vasculature with PBS to eliminate contaminating blood from the brain. Resultant brain tissues were homogenized in PBS and CCS1477 quantitated with a qualified LC-MS/MS method. At these timepoints and doses, we were unable to quantitate CCS1477 in brain tissue with acceptable precision and accuracy (i.e. results were below the lower limit of quantitation of 6 ng/mL). These data indicated that MB cells, and G3MB cell lines in particular, may display enhanced sensitivity to BRD-based inhibition of EP300/CBP, as compared with HAT domain inhibition, though the ability of CCS1477 to target EP300/CBP in vivo is limited by poor blood-brain barrier penetration.

## CCS1477 preferentially targets EP300 and CBP

G3MB cells were exceptionally sensitive to CCS1477, as compared to A485, though this compound appeared to be non-blood-brain barrier (BBB) penetrant. Thus, we hypothesized that a thorough dissection of compound selectivity and activity may allow for compound refinement, with an ultimate goal of improving BBB penetration and retaining on-target activity against EP300/CBP. The BRDs of EP300 and CBP are highly structurally related to each other[10]. Further, the BRDs of EP300 and CBP have structural similarity with minor sequence homology to those found in other proteins, including the BD1 N-terminal bromodomains of the BET proteins BRD2,3,4 and BRDT (Supplementary Fig. 3a). Prior studies have demonstrated strong efficacy of BRD4 inhibitors, such as JQ1, in G3MB cells in vitro[47–51]. Thus, to ensure that the effects we observed on suppression of G3MB cell growth were specific to inhibition of the EP300/CBP BRD and not due to cross-reactivity with other BRD-containing proteins, we tested the effects of CCS1477 by Bromoscan profiling across 32 human BRDs. We observed strong binding of CCS1477 to the BRD of CBP and EP300, but also with the BD1 BRDs of BRD2, 3 and 4 (Fig. 2a, Supplementary Table 3). These data were contrasted with published $K_d$ values derived from surface plasmon resonance measurements of 1.3, 1.7 and 222 nmol/L for EP300, CBP and BRD4, respectively[5]. To resolve these findings, we performed binding studies using differential scanning fluorimetry (DSF), to assess the binding of CCS1477 with recombinant BRDs from EP300, CBP or the first bromodomain (BD1) of BRD4 (Fig. 2b). CCS1477 produced a greater thermal shift with EP300 or CBP, as compared with BRD4. Similar observations, though much less potent, were seen using the parental compound for CCS1477, SGC-CBP30 (Fig. 2b). Further analysis of the binding energies by isothermal titration calorimetry (ITC) confirmed significantly stronger interaction of CCS1477 with the bromodomains of CBP and EP300 ($K_d$ = 4.0 and 26 nM, respectively) than with BD1 of BRD4 ($K_d$ = 403 nM) (Fig. 2c, Table 1). Thus, CCS1477 displays preferential binding to the bromodomains of EP300 and CBP, compared with BRD4.

To understand the structural basis for the observed differential binding affinities of CCS1477 between EP300/CBP and BRD4, we next solved the co-crystal structures of the respective bromodomains with CCS1477 (Fig. 2d). As expected, CCS1477 binds through canonical hydrogen-bonding interactions of its dimethyl isoxazole moiety with a conserved asparagine residue in the acetyl-lysine (Kac) binding site (CBP[N1168], EP300[N1132], BRD4[N140]). A major difference, however, between the inhibitor binding pattern in EP300/CBP and BRD4 is the involvement of an arginine side chain (EP300[R1137], CBP[R1173]) that interacts with the difluorophenyl ring of CCS1477 through Pi-cation interactions, while establishing an additional hydrogen-bond with the piperidinone oxygen (Fig. 2e). BRD4 is devoid of an arginine in this region of the Kac site, which explains the reduced binding affinity for CCS1477. Additionally, the BRD4 binding pocket contains a WPF shelf that imposes steric hindrance on CCS1477 such that the side chain of BRD4[W81] undergoes a conformational change to accommodate the difluorophenyl moiety (Supplementary Fig. 3b). In EP300/CBP, an analogous tryptophan residue is not present, and instead the equivalent LPF shelf is less bulky and facilitates interaction with CCS1477. Superposition of CCS1477 as bound in the respective Kac sites reveals an altered binding pose, reflecting the reduced shape complementarity between CCS1477 and BRD4, as compared with EP300/CBP (Fig. 2f).

Building on these structural observations, we next identified whether CCS1477 interacted preferentially with EP300, CBP or BRD4 in in vitro biochemical assays. Since the methoxycyclohexyl functional group of CCS1477 does not display significant interactions or steric hindrance, we used this group as an ideal location for biotin conjugation. With this probe, we performed biotinylated pulldown assays in cell extracts derived from HDMB03 G3MB and Kelly neuroblastoma cells (Fig. 2g, Supplementary Fig. 3c). Lysates were treated with biotinylated-CCS1477 at 1 or 10 μM, prior to western blotting. These data demonstrated that at lower concentrations, biotinylated-CCS1477 interacted with EP300 but not BRD4 (Fig. 2g, Supplementary Fig. 3c). As a control, this interaction could be ablated by co-incubation with a 10-fold excess of unlabeled CCS1477 (Fig. 2g, Supplementary Fig. 3c). At supraphysiologic high doses (10 μM), biotin-CCS1477 interacted

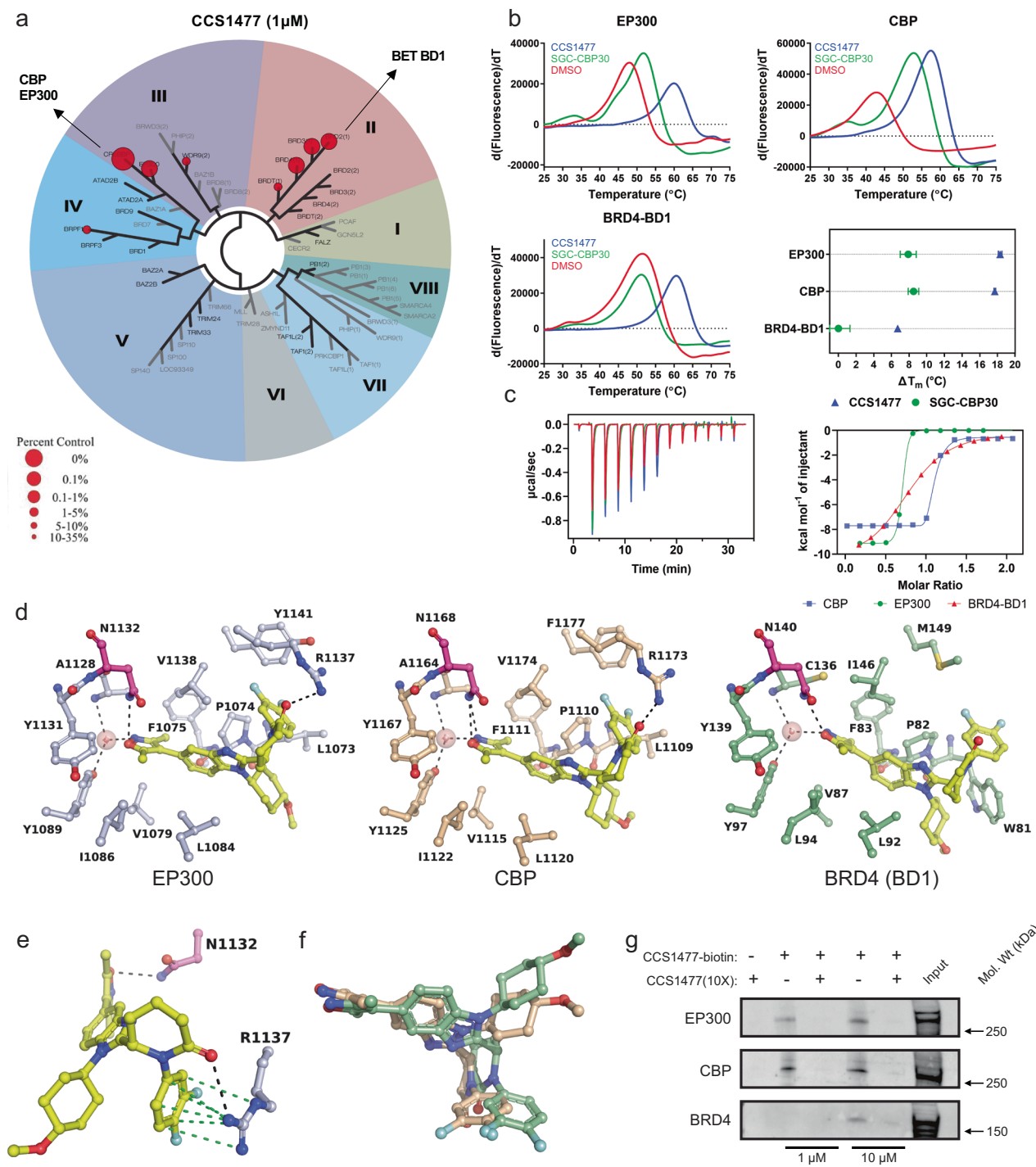

with EP300 and BRD4 (Fig. 2g, Supplementary Fig. 3c). Together, these data demonstrate that CCS1477 is a potent inhibitor of EP300/CBP and a moderate inhibitor of the first bromodomain of BRD4 and other BET proteins. At concentrations relevant to inhibition in cells (≤1 μM) CCS1477 likely predominantly interacts with EP300 and CBP.

**Structure-activity relationship (SAR) studies of the CCS1477 pharmacophore**
Next, we sought to use this baseline structural resolution of the EP300 and CBP BRDs to develop an understanding of the potency and selectivity of CCS1477. To further understand the selectivity of CCS1477 for EP300 and CBP over BRD4, we generated a set of CCS1477 derivatives. First, we tested the hypothesis that the difluorophenyl

moiety of CCS1477 ensures efficient interactions with the BRDs of EP300 or CBP (Fig. 2e). To do so, we synthesized analogues of CCS1477 and characterized them for binding affinity by DSF, microscale thermophoresis (MST) and ITC (Table 1). The $\Delta T_m$ values from DSF studies strongly correlated for compound interactions with EP300 and CBP (Fig. 3a, b), reflecting the high similarity of the Kac site between these paralogs. The correlation of DSF values for CBP and BRD4 was less significant, indicating differences for compound interactions with the respective Kac sites (Supplementary Fig. 3d). This is also evident from the ratio of $\Delta T_m$ values for CBP/BRD4, which indicates changes in selectivity of certain compounds for CBP over BRD4 (Supplementary Fig. 3e). Since recombinant CBP BRD was more robust in biochemical assays and crystallization studies, $K_d$ values were determined for all

**Fig. 2 | CCS1477 preferentially targets the bromodomains of EP300 and CBP.**
**a** Profiling of CCS1477 (1 μM) against human bromodomains demonstrates pre-
ferential interaction of CCS1477 with CBP and EP300, and additional interactions
with BD1 of the BET proteins BRD2, BRD3, BRD4 and BRDT (BromoScan by Dis-
coverX). Tabulated data are shown in Supplementary Table 3. **b** Representative
melting curves from DSF studies of the bromodomains of EP300, CBP and BRD4-
BD1 in the presence of CCS1477 or SGC-CBP30 along with a summary graph ($n = 3$,
error bars represent standard deviation, SD). Source data are provided in Source
Data file. **c** Isothermal titration calorimetry (ITC) analysis of the interaction of
CCS1477 with the bromodomains of EP300, CBP and BRD4-BD1; $K_d$ values were
$25.5 ± 46.3$, $4.0 ± 6.7$, and $403 ± 137$ nM, respectively ($n = 1$; error represents stan-
dard error of mean of technical replicates, SEM). See also Table 1. Source data are
provided in Source Data file. **d** Cocrystal structures of CCS1477 (yellow) bound to
CBP (beige, PDB code 8FV2), EP300 (grey, PDB code 8FVF), or BRD4-BD1 (green,

PDB code 8FVK). Black dotted lines indicate hydrogen bonding interactions, the
critical asparagine residue is highlighted in magenta, water molecules are shown as
pink spheres. Crystallographic data and refinement statistics are in Supplementary
Tables 5 and 6, the electron density maps of bound inhibitor are in Supplementary
Fig. 4. **e** Mixed H-bonding and Pi-cation interactions (green dotted lines) between
the side chain of $R^{1137}$ and the difluorophenyl-piperidone moiety of CCS1477 in
EP300. The same interaction pattern is seen in CBP with $R^{1173}$, while BRD4-1 lacks an
equivalent arginine residue in this region of the binding site. **f** CCS1477 adopts
different conformational states in CBP (beige) and BRD4-BD1 (green), reflecting
differences in shape complementarity with the respective KAc sites. **g** Biotinylated-
CCS1477 pulldowns in HDMB03 cell lysates demonstrates pulldown of EP300 and
CBP, but not BRD4 at low concentrations, and interaction with EP300, CBP and
BRD4 at higher concentrations of compound. Data is representative of $n = 3$ inde-
pendent lysates and reactions.

compound-CBP interactions by MST, and co-crystal structures of the
CBP BRD with five of the seven analogues were determined. We
observed that the $\Delta T_m$ values from DSF strongly correlated with the $K_d$
values from MST, which confirmed the robustness of binding affinity
assessment by the chosen orthogonal assays (Fig. 3c).

To directly test the contribution of the difluorophenyl moiety, we
generated derivatives lacking the difluorophenyl portion (compounds
**1** and **2**, structures for all derivative compounds detailed in Table 1).
The resultant compounds displayed a >1000-fold loss of affinity for
EP300 and CBP (Table 1). A co-crystal structure of the CBP bromodo-
main with compound **1** demonstrated that the side chain $R^{1173}$ is more
flexible than in co-crystal structures with CCS1477 and assumes a
conformation incompatible for hydrogen-bonding with the piper-
idinone oxygen, while weakly interacting with a nitrogen of the ben-
zimidazole core (Fig. 3d). Therefore, loss of Pi-cation as well as
hydrogen-bonding interactions renders compounds **1** and **2** less effi-
cient binders to EP300 and CBP. Notably, activity against BRD4 was
reduced by ~10-fold, indicating that the hydrophobic van-der-Waals
(VDW) interactions observed between the difluorophenyl moiety of
CCS1477 and the WPF shelf of BRD4 contribute to binding affinity, but
relatively less so in the EP300/CBP bromodomain. Introducing bis-
trifluoromethyl in *meta* position of the phenyl ring (compound **4**)
reduced the binding affinity by ~40-fold; while the co-crystal structure
revealed Pi-cation and hydrogen-bonding interactions with $R^{1173}$, steric
hindrance imposed on the LPF shelf leads to conformational changes
that negatively impact overall binding affinity (Fig. 3e). Substitution of
difluorophenyl with a less bulky phenyl (compound **5**) resulted in a ~10-
fold loss of binding activity, and the cocrystal revealed that $R^{1173}$ is now
hydrogen-bonded to the piperidinone oxygen but not quite close
enough to the phenyl ring to establish a Pi-cation interaction (Fig. 3f).
Introduction of a butyl group in *meta* position of the phenyl (com-
pound **6**) did not change the binding affinity relative to **5**, and the
cocrystal structure with CBP showed that $R^{1173}$ establishes the same Pi-
cation and hydrogen-bonding interactions as seen with CCS1477
(Fig. 3g). Notably, replacement with a dihydrobenzodioxine moiety
(compound **7**) maintained high binding affinity ($K_d = 17$ nM), and the
cocrystal structure confirmed unimpeded positioning and interaction
with $R^{1173}$ (Fig. 3h). Illustrating these SAR findings, electron density
maps of ligand binding and binding poses of ligand-protein pairs are
found in Supplementary Figs. 4 and 5. While none of the analogues
were superior to CCS1477 in terms of binding affinity, substitutions
were well tolerated if productive interactions with $R^{1173}$ were main-
tained. Among the analogues tested, only compound **6** maintained
high activity against CBP but was considerably less potent against
BRD4, indicating an increase in target selectivity. Correspondingly,
while these compounds displayed increased $IC_{50}$ values in HDMB03
G3MB cells (Table 1), they also demonstrated enhanced selectivity for
the BRD of CBP, compared with BRD4 (Supplementary Fig. 3e), sug-
gesting more on-target activity against EP300/CBP. Combined, these
SAR studies suggest that modifications of the CCS1477 parent

compound to increase efficacy for certain applications, such as
improved target selectivity or facilitating BBB penetrance, may be
feasible.

These biochemical data were concordant with in vitro pulldown
binding assays using biotinylated CCS1477-int(1), where, in contrast to
biotinylated CCS1477, high doses failed to significantly interact with
EP300, CBP or BRD4 (Fig. 3i). These structural studies were further
emphasized by in vitro testing in G3MB cells, where loss of the
difluorophenyl moiety, resulted in ablated binding to EP300/CBP
(compounds **1, 2**) and caused a drastic decrease in medulloblastoma
cell growth inhibition (Table 1). In contrast, milder changes to com-
pound structure that retain the ability of compound to form Pi-cation
as well as hydrogen-bonding interactions with the bromodomain
result in only minor changes in $IC_{50}$ (Table 1). To further evaluate the
specific effects of CCS1477-int(1) as a case example, we performed
further CellTiter-Glo dose-response growth assays in HDMB03 and
MB002 G3MB cells, demonstrating a blunting of anti-growth effect in
cells treated with CCS1477-int(1), as compared with CCS1477 (Fig. 3j, k).
Thus, the difluorophenyl group is a critical moiety in the CCS1477
pharmacophore that facilitates potent interaction with the bromodo-
main of EP300/CBP and is required for efficacy in cell line models
of G3MB.

**The bromodomain of EP300/CBP is required to maintain tran-
scriptional dependency networks in Group 3 medulloblastoma**
Since EP300/CBP are dominant regulators of gene expression, next we
determined the direct, early effects of CCS1477 treatment of G3MB
cells in vitro. We treated either HDMB03 or MB002 cells with the day 3
$IC_{50}$ dose of CCS1477 for a short time-period of 6 h, and then extracted
total cellular RNA for ERCC-controlled spike-in RNAseq analysis. As a
comparison, we treated cells in parallel with the day 3 $IC_{50}$ dose of
A485 or the BRD4 inhibitor, JQ1 for the same timepoints. First, we
examined global changes in gene expression induced by drug treat-
ment. Independently, for each cell line and condition, we determined
significantly differentially expressed genes using DEseq2, relative to
DMSO treated controls (Supplementary Fig. 6a). By examining the
expression of any significantly altered gene across all treatments and
cell lines, relative to DMSO controls, we observed a distinct modular
pattern of gene expression changes induced by CCS1477, A485 or JQ1
(Fig. 4a). Next, we identified high-confidence genes showing sig-
nificantly and coordinately changed expression by treatment with
each compound, in both cell lines. More genes showing significant
changes were downregulated than upregulated, with A485 demon-
strating the fewest effects, followed by CCS1477, and then, consistent
with prior reports, JQ1 (Fig. 4b; Supplementary Fig. 6a).

Since CCS1477 and A485 demonstrated dramatically different
effects on G3MB cell growth (Fig. 1g, h), and different patterns of gene
dysregulation (Fig. 4b), we next sought to identify pathways that were
dysregulated by domain-specific compound treatment in HDMB03
and MB002 G3MB cells. We compared the relative effects of A485 and

## Table 1 | Characterization of CCS1477 and derivatives

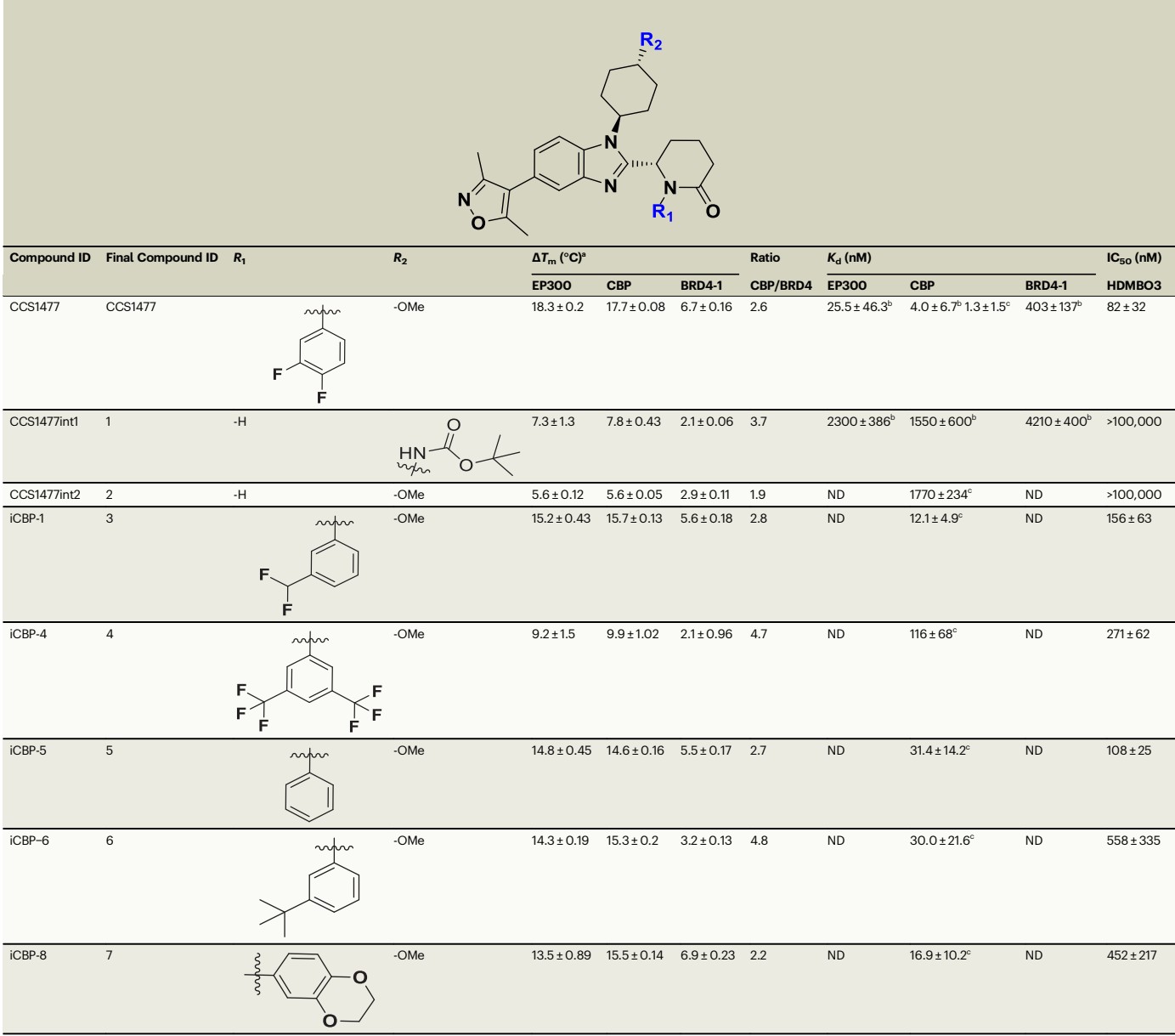

| Compound ID | Final Compound ID | $R_1$ | $R_2$ | $\Delta T_m$ (°C)[a] | | | Ratio | $K_d$ (nM) | | | IC$_{50}$ (nM) |
| --- | --- | --- | --- | --- | --- | --- | --- | --- | --- | --- | --- |
| | | | | EP300 | CBP | BRD4-1 | CBP/BRD4 | EP300 | CBP | BRD4-1 | HDMB03 |
| CCS1477 | CCS1477 | | -OMe | 18.3 ± 0.2 | 17.7 ± 0.08 | 6.7 ± 0.16 | 2.6 | 25.5 ± 46.3[b] | 4.0 ± 6.7[b] 1.3 ± 1.5[c] | 403 ± 137[b] | 82 ± 32 |
| CCS1477int1 | 1 | -H | | 7.3 ± 1.3 | 7.8 ± 0.43 | 2.1 ± 0.06 | 3.7 | 2300 ± 386[b] | 1550 ± 600[b] | 4210 ± 400[b] | >100,000 |
| CCS1477int2 | 2 | -H | -OMe | 5.6 ± 0.12 | 5.6 ± 0.05 | 2.9 ± 0.11 | 1.9 | ND | 1770 ± 234[c] | ND | >100,000 |
| iCBP-1 | 3 | | -OMe | 15.2 ± 0.43 | 15.7 ± 0.13 | 5.6 ± 0.18 | 2.8 | ND | 12.1 ± 4.9[c] | ND | 156 ± 63 |
| iCBP-4 | 4 | | -OMe | 9.2 ± 1.5 | 9.9 ± 1.02 | 2.1 ± 0.96 | 4.7 | ND | 116 ± 68[c] | ND | 271 ± 62 |
| iCBP-5 | 5 | | -OMe | 14.8 ± 0.45 | 14.6 ± 0.16 | 5.5 ± 0.17 | 2.7 | ND | 31.4 ± 14.2[c] | ND | 108 ± 25 |
| iCBP−6 | 6 | | -OMe | 14.3 ± 0.19 | 15.3 ± 0.2 | 3.2 ± 0.13 | 4.8 | ND | 30.0 ± 21.6[c] | ND | 558 ± 335 |
| iCBP-8 | 7 | | -OMe | 13.5 ± 0.89 | 15.5 ± 0.14 | 6.9 ± 0.23 | 2.2 | ND | 16.9 ± 10.2[c] | ND | 452 ± 217 |

CCS1477, CCS1477 intermediates and derivative compounds were characterized by differential scanning fluorescence (DSF), microscale thermopheresis (MST) and isothermal calorimetry to determine compound characteristics, using recombinant bromodomains from EP300, CBP or BD1 of BRD4 (BRD4-1).
[a]Three independent DSF experiments (average ± SD).
[b]Single ITC experiment (data fit ± SEM).
[c]Three independent MST experiments (average ± SD).

CCS1477 treatment on HDMB03 and MB002 cells by Gene Set Enrichment Analysis (GSEA) using the Hallmarks genesets from the Molecular Signatures Database (MSigDB). The most consistently altered Hallmark geneset between CCS1477 and A485-treated cells was the MYC_Targets_V2 geneset, which was collectively downregulated in both cell lines when treated with CCS1477, as compared with A485 (Fig. 4c; Supplementary Fig. 6b). This observation was intriguing given that both G3MB cell lines harbor high-level amplification of *c-MYC*, which is a known oncogene in the disease[39,46,50,52]. Concordant with these findings, we observed loss of *MYC* gene expression and protein levels in HDMB03 cells treated with CCS1477, and to a lesser extent with A485 and JQ1, at this timepoint (Fig. 4d; Supplementary Fig. 6c). Despite the observation of moderate loss of *c-MYC* expression with A485, this was enhanced with CCS1477 treatment (Fig. 4c, d; Supplementary Fig. 6b, c).

Next, we sought to identify additional linked genes and pathways responsible for these disparate growth effects. We performed a Metascape analysis of the genes significantly downregulated by CCS1477, as compared with DMSO, focusing on the oncogenic signatures dataset from MSigDB. Filtering this for significant enrichment ($\log_{10}$Q-value < −3), we identified an enrichment of MYC-regulated genesets, in addition to several others (Fig. 4e). Similar analysis comparing A485 or JQ1 against DMSO demonstrated enrichment for different oncogenic signatures, including TP53 and E2F3 signatures for JQ1-treated cells, and MEK signatures for A485-treated cells (Fig. 4e). Since MYC-regulated genesets appeared to be related to the CCS1477 effect, next we examined the promoters of genes downregulated by CCS1477, A485 or JQ1 for MYC consensus binding sequences. Concordant with the effects on Hallmarks and oncogenic signatures, we identified enrichment of MYC binding motifs for CCS1477-

downregulated genes, which was less prominent in A485- or JQ1-downregulated genesets (Supplementary Fig. 6d). These findings suggested a key enhanced, but perhaps not sole role for c-MYC in driving G3MB early responses to EP300/CBP BRD inhibitor treatment, compared with HAT inhibitor treatment.

Next, we sought to identify the early gene networks that are disrupted by each individual inhibitor treatment that were functionally responsible for reduced HDMB03 cell growth. To do so, we examined genes that were significantly downregulated by each compound and intersected these data with orthogonal exome-wide CRISPR-cas9 dropout screening data from seven MB cell lines[35]. We observed that 19.4% of genes significantly downregulated by CCS1477 were required for growth of MB cells, which was higher than that found after treatment with JQ1 (12.5%) or A485 (8.6%) ($p = 0.0027$ (A485 vs. CCS1477) and $p = 0.0034$ (JQ1 vs. CCS1477) by two-sided Fisher's exact test, Supplementary Fig. 6e). Gene ontology analysis using the PANTHER tool[53] demonstrated that >50% of genes downregulated by CCS1477 and required for MB growth were nucleic-acid-binding proteins, transcription factors or chromatin-binding proteins (Fig. 4f). In contrast, other ontologies were enriched in A485 and JQ1 treatment (Supplementary Table 4). Given this, we next sought to identify if inhibitor treatment affected a central process. We used the STRING database[54] to determine whether the functionally relevant genes downregulated by CCS1477, A485 or JQ1 formed a candidate protein-protein interaction network. This analysis demonstrated that the majority of genes required for MB cell growth and disrupted by CCS1477 were involved in a highly interconnected and integrated protein-protein interaction network involved in coordination of mRNA transcription/cell cycle regulation (red) or RNA metabolism (blue) (Fig. 4g, terms determined by Gene Ontology analysis). In contrast, the dependency genes disrupted by A485 produced proteins that were largely unlinked from each other, and not enriched for a specific functional category by GO analysis (Supplementary Fig. 6f denoted by lack of interconnections between proteins, Supplementary Table 4). As a control, JQ1 disrupted several small networks of proteins whose genes are required for growth of MB cells, though these networks were far less interconnected than those targeted by CCS1477 (Supplementary Fig 6g; Supplementary Table 4). These findings indicate that the primary effect of CCS1477 on transcription in HDMB03 cells is through dysregulation of a network of genes involved in RNA metabolism and coordinated by MYC proteins, that is critically required for MB cell growth.

## Discussion

Epigenetic dysregulation of transcription is common in cancer cells, resulting in transcriptional amplification, developmental arrest and oncogenesis[55]. These processes are mediated by cohorts of transcription factors and coregulators, which cumulatively result in a cancer phenotype[55,56]. Broad efforts to target these dysregulated processes have yielded agents that can suppress transcription (reviewed in[1]). Significant efforts have led to the development of compounds useful to target epigenetic reader, writer and eraser proteins, however these proteins often contain multiple functional domains with unique inhibitors designed to each[1]. The choice of which inhibitor, for which protein, to achieve optimal cellular effect, remains a critical problem for drug discovery and eventual clinical translation.

EP300 and CBP are paralogous, dominant epigenome-regulating enzymes associated with control of transcription. These multidomain lysine acetyltransferases are associated with active chromatin and acetylate susceptible lysine residues of a variety of target proteins, resulting in altered protein localization, stability and function[8]. Classical histone targets of these proteins include marks of active chromatin regions, including gene enhancers and promoters[57–59], through which these proteins promote RNA polymerase and complex assembly and transcriptional activity[9]. High concentrations of EP300/CBP

proteins promote phase separation at super-enhancer elements, which promotes high-level transcription of target genes[60]. These functions are enabled by the bromodomains of EP300 and CBP, through which they bind to acetylated proteins[5,10,11,32]. These catalytic and binding-based effects are linked with transcriptional fidelity, since combined HAT and bromodomain targeting of EP300/CBP or targeted degradation of full-length proteins results in synergistic loss of transcription[16,19,31,32].

Much like many other epigenetic regulatory proteins[1], significant medicinal chemistry efforts have led to the production of a variety of compounds targeting distinct domains within EP300 and CBP proteins, in addition to newer proteolysis-targeted chimaera molecules which induce the degradation of one or the other proteins[1,5,11,16,21,22,26]. All currently available small molecule inhibitors are nonselective between EP300 and CBP, resulting in toxicity due to cross-reactivity when targeting these critical proteins. Regardless, the central role of these proteins in driving tumorigenesis through regulation of oncogenic transcription has led to significant pharmaceutical interest in advancing seed molecules into clinical trials.

Recognizing this concern regarding toxicity, we sought to identify whether different cancers show enhanced susceptibility to bromodomain or HAT domain inhibition, for the purposes of nominating specific tumors for further exploration of specific types of EP300/CBP inhibitors. Here, we used a chemical-based high-throughput functional screen to identify specific malignancies, based on cell line data, in which specific domain inhibition had exceptional effects. Previous studies using this screening method demonstrated that experimental repeats conducted with the same compounds[33] or similar compounds affecting the same target[34] are highly correlated. Further, low throughput confirmation of selected cell lines in our experiments confirmed our screening results (Fig. 1b–e, Supplementary Fig. 1e, f). In comparing two distinct compounds in high-throughput screening, however, variables that are challenging to control include variability in compound-cell permeability between different cell lines, and the reliance of some cell lines, notably, leukemic cell lines, on secreted factors for survival. Despite these concerns, we observed that, while HAT inhibition was superior to bromodomain inhibition in B-lineage acute lymphoblastic leukemias, several tumor types, including endometrial carcinoma, multiple myeloma and medulloblastoma demonstrated enhanced effects with EP300/CBP bromodomain inhibition. These results are consistent with recent publications demonstrating CCS1477-based inactivation of distinct oncogenic networks in multiple myeloma[37]. Importantly, the majority of cancer cells are similarly sensitive to targeting EP300/CBP proteins through chemical inhibition of either the histone acetyltransferase or bromodomains of these proteins. These findings indicate a potential key strategy for target development, in addition to highlighting key issues of toxicity and pharmacokinetic profiles in compound development, where compounds with improved stability, permeability and solubility would be favored.

We observed that bromodomain inhibition demonstrated exceptional effects in medulloblastoma cell lines. Analysis of mutation, expression, methylation and dependency data from the Cancer Cell Line Encyclopedia[61] failed to identify a correlation between mutational status of EP300/CBP and sensitivity to either HAT or bromodomain inhibition, including strong relationships to *MYC* genes (Supplementary Fig. 1). These data indicate that *MYC* expression may be only one component of the response to these compounds. This concept is echoed by our findings that extensive networks of dependencies are lost in G3MB cells after treatment with CCS1477, compared with A485, which include *MYC* but also other dependency genes. Further, these data are supported by studies demonstrating that while CCS1477 inactivates *MYC* in prostatic carcinoma[5], other tumors such as acute myeloid leukemia and multiple myeloma have distinct oncogenes affected by CCS1477 treatment[37].

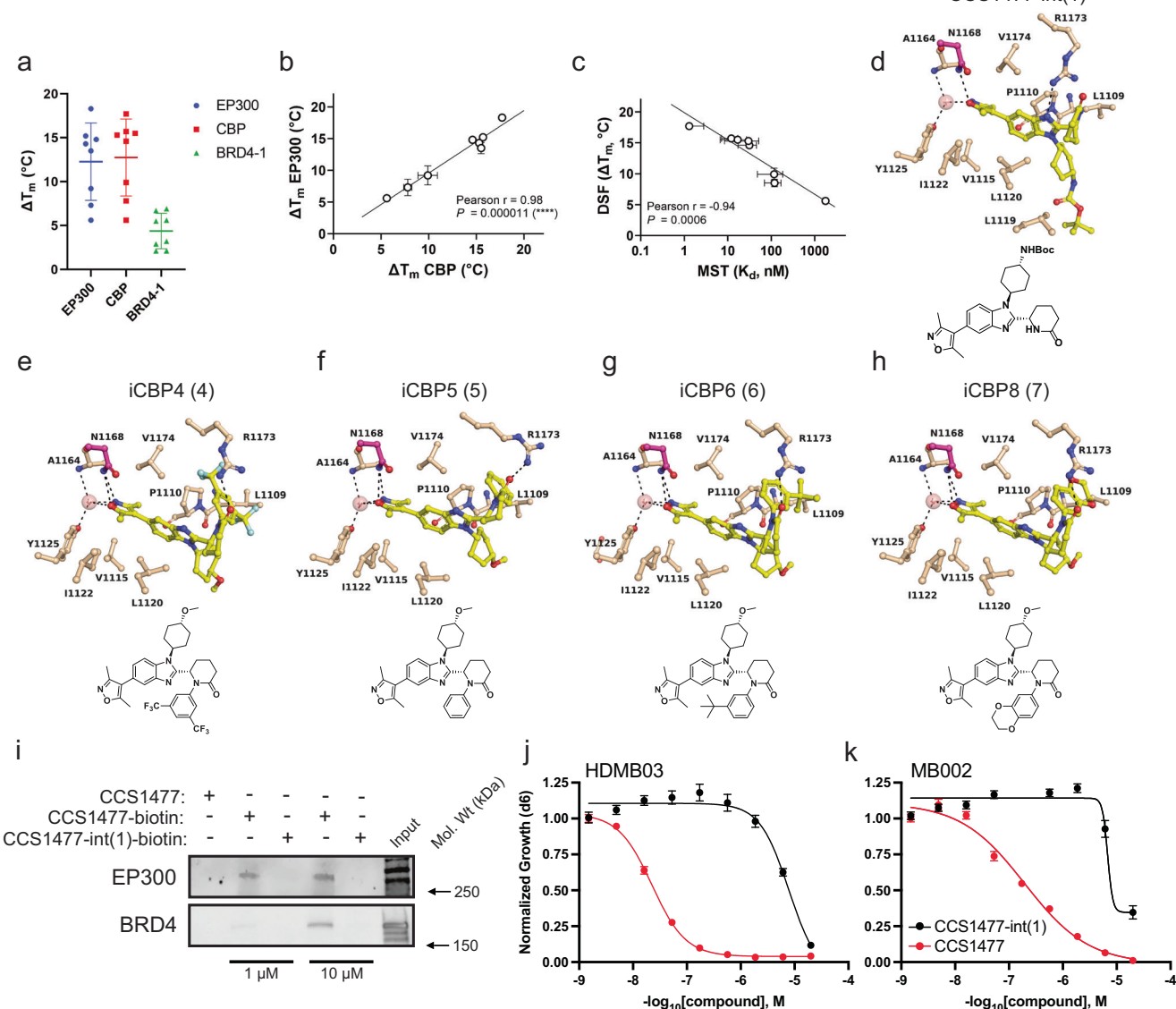

**Fig. 3 | Structure-activity relationship (SAR) studies of CCS1477 analogues reveal the importance of compound engagement with R$^{II73}$ (CBP) or R$^{II37}$ (EP300) for inhibitory activity. a** Graphical representation of the binding affinity of CCS1477 and analogues thereof for CBP, EP300 and BRD4-BD1 as assessed by DSF. Bars represent the mean and standard deviation for each protein across all inhibitors ($n = 3$). Source data are provided in Source Data file. **b** Correlation of $\Delta T_m$ values for the bromodomains of CBP and EP300. The Pearson's $r$ and statistical significance $P$ values (two-tailed) are indicated. Error bars represent SD of each DSF experiment ($n = 3$). Source data are provided in Source Data file. **c** Correlation of $\Delta T_m$ and $K_d$ values from MST experiments for compound interaction with CBP. The Pearson's $r$ and statistical significance $p$ values (two-tailed) are indicated. Error bars represent the standard deviation of each DSF ($n = 3$) and MST ($n = 3$) experiment. Tabulated $\Delta T_m$ and $K_d$ values are shown in Table 1. Source data are provided in

Source Data file. **d**–**h** Co-crystal structures of CBP with compounds **1** (PDB 8FVS), **4** (PDB 8FXA), **5** (PDB 8GA2), **6** (PDB 8FXE), and **7** (PDB 8FXO). Compound structures are shown below co-crystal structures. 2D diagrams of the binding interactions are shown in Supplementary Fig. 5. **i** Biotinylated-CCS1477 pulldowns in HDMBO3 medulloblastoma cell lysates demonstrates pulldown of EP300 but not BRD4 at low concentrations of CCS1477, and interaction with EP300 and BRD4 at higher concentrations of CCS1477. CCS1477-int1 fails to interact with either EP300 or BRD4 at either concentration. Data is representative of $n = 3$ independent lysates and reactions. **j**, **k** HDMBO3 (**j**) and MB002 (**k**) cells were tested for dose-response effects of CCS1477 and CCS1477-int1 after 6 days by Cell-Titer Glo assay. $n = 3$ independent biological replicates for each dose. Error bars represent S.E.M. CCS1477 data as demonstrated in Fig. 1g, h. Source data are provided in Source Data file.

Building on these results, mutation of EP300 or CBP is found in nearly 5% of medulloblastoma tumors[40]. Intriguingly, analysis of COSMIC[62] and the St. Jude Children's Research Institute ProteinPaint[63] resources indicates that while missense mutations are spread across the coding exons of EP300 or CBP, including the HAT domain, the bromodomain of either *EP300* or *CBP* is not mutated. These data are striking, considering that bromodomain mutations are found in EP300 or CBP in other tumors[62], and may suggest that the intact bromodomain, and therefore, scaffolding activity of EP300 and CBP, is of particular importance to the biology of these tumors. This hypothesis is

consistent with our findings that EP300/CBP bromodomain inhibition has enhanced effects on these tumor cells. Further large-scale sequencing efforts are required to correlate these observations and determine the drivers of enhanced reliance on individual subdomains in medulloblastoma and other distinct tumor states, which may be variable.

To understand the modes of binding and specificity of CCS1477 for EP300, CBP and BRD4, we performed crystallographic experiments, combined with modifications of the CCS1477 molecule, to develop a structure-activity relationship of the catalytic core of

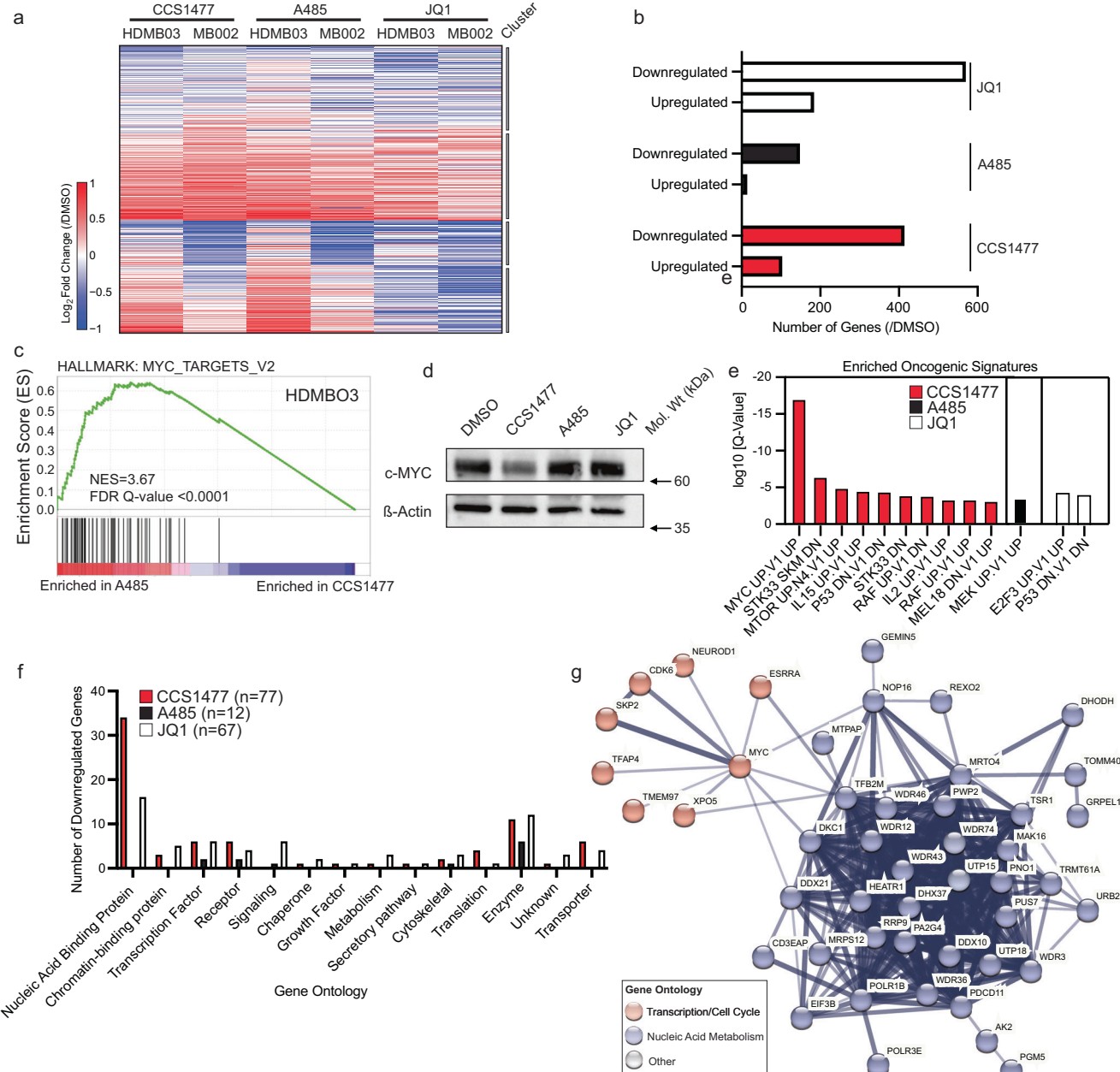

**Fig. 4 | CCS1477 selectively disrupts an interlinked transcriptional genetic dependency network in group 3 medulloblastoma cells including *MYC*.**
**a** HDMBO3 and MB002 cells were treated with d3 IC$_{50}$ of CCS1477 (HDMBO3 = 39 nM, MB002 = 1280 nM), A485 (HDMBO3 = 400 nM, MB002 = 1970nM), JQ1 (HDMBO3 = 100 nM, MB002 = 107 nM) or matched concentration of DMSO control for 6 h, followed by ERCC-controlled spike-in RNAseq analysis. DESEQ2 analysis was performed to detect significantly changed transcripts (adjusted $p < 0.05$ comparing treatment and DMSO). Transcripts that are significant in any treatment or cell line are shown ($n = 5092$). Heatmap displays k-means ranked clusters of significantly different genes found in HDMBO3 and MBOO2 cells ($k = 4$, grey bars indicate clusters). **b** Union of statistically significant up and downregulated transcripts, relative to DMSO controls from (**a**) found in both HDMB03 and MB002 cells. $n = 3$ independent biological replicates per treatment. **c** Gene set enrichment analysis comparing ERCC spike-in normalized transcriptomes of A485 or CCS1477 treated HDMBO3 samples, using the Hallmarks dataset identified the top

differentially regulated gene set to be "Hallmarks_MYC_Targets_V2" with an NES = 3.67 and FDR $q$-value of 0. **d** HDMBO3 cells were treated with A485 (400 nM), CCS1477 (39 nM) or JQ1 (100 nM) for 6 h followed by lysis for western blotting for c-MYC proteins. β-actin is demonstrated as a loading control. Data is representative of three independent biological replicates and blots. **e** Metascape analysis using the MSigDB gene set "Oncogenic Signatures" database, demonstrating the log$_{10}$ $q$-value of oncogenic signatures lost in genes downregulated by either CCS1477, A485 or JQ1 treatment. **f** Gene annotations from PANTHER analysis of genetic dependencies in 7 medulloblastoma cell lines that are downregulated by treatment with CCS1477, A485 or JQ1. **g** String-database analysis of high-stringency CCS1477-downregulated gene dependencies in medulloblastoma cell lines demonstrates a highly interlinked dependency network centered on transcriptional regulation. Line width indicates strength of known protein-protein interactions. Red indicates proteins involved in control of mRNA transcription/cell cycle, blue indicates proteins involved in RNA metabolism.

CCS1477. Our findings demonstrate key residue interactions in the bromodomain of EP300/CBP and CCS1477 (Figs. 2 and 3). Chemical substitution of the difluorophenyl ring of CCS1477 yielded a compound with drastically reduced inhibition and poor binding to either EP300 or CBP, highlighting a critical functional role for this moiety within the catalytic core of the CCS1477 molecule. These studies illuminating the pharmacophore of CCS1477 are crucial to the development of derivatives that maintain on-target activity, with evolved characteristics, such as improved BBB penetration, for use in high-risk medulloblastoma.

Prior evidence has implicated a key role for BRD4 inhibition in multiple tumor states, including G3MB[47,48,51,64] among many others, where dominant phenotypic outcomes on transcription were related to disruption of *MYC* transcription. Of note, these studies typically have used longer treatments (>24 h) of JQ1 or related molecules to elicit effects on *MYC* transcription. In our study, we observed a higher statistical enrichment of *MYC* binding motifs in the promoters of genes downregulated by CCS1477, compared with those downregulated by A485 or JQ1 (Supplementary Fig. 6d). In combination with observations that CCS1477-treatment induces downregulation of MYC protein by western blotting (Fig. 4d), our observations suggest that CCS1477 more rapidly and potently causes disruption of *MYC* mRNA levels than do A485 or JQ1.

Here, we demonstrate two key observations: (1) Using biochemical, structural and medicinal chemistry approaches, CCS1477 has enhanced specificity for the bromodomains of EP300 and CBP relative to BD1 of BRD4; and (2) The early effects of CCS1477, but not A485 or JQ1 treatment, in G3MB cell lines involves selective disruption of networks of genes required for survival of these cells, including *c-MYC* as previously reported[5], in addition to networks of critical genes beyond *c-MYC* as well. Our observations highlight that inhibition of EP300/CBP bromodomains function beyond regulation of *c-MYC* to include dysregulation of gene networks involved in regulation of transcription and RNA splicing including sequence-specific transcription factors, RNA synthesis, transport and processing genes, and cell cycle genes, many of which are previously known to be linked to MB growth[40,65]. These findings were not observed with HAT domain inhibitors, which demonstrated striking differences with BRD inhibition. Despite having similar outcomes in cell growth at day 3, the transcriptional effects of JQ1 and A485 were strikingly different from those of CCS1477. These differences may reflect either the kinetics by which CCS1477 induces transcriptional dysregulation in G3MB cells, as compared with A485 or JQ1, or the direct targets that are associated with bromodomain or HAT domain activity. At this timepoint, treatment with A485 or JQ1 both induce transcriptional loss of *MYC*, though are insufficient to cause loss at the protein level. Therefore, ongoing studies are aimed at understanding if more prolonged treatment with A485 or JQ1 is sufficient to inactivate similar networks of dependency genes involved in transcription, as are achieved by rapid treatment with CCS1477. Despite this, these studies indicate that domain-specific inhibition may be a general property for consideration in the derivation of targeted therapeutics. These results echo the distinct phenotypes elicited by targeting different bromodomains of BET proteins[64,66,67].

In summary, our findings indicate that there are distinct effects of targeting individual subdomains of EP300/CBP in a restricted range of tumors. In particular, the use of EP300/CBP bromodomain inhibitors in high-risk tumors such as Group 3 medulloblastoma is of interest, due to rapid and selective effects on disrupting networks of transcriptional regulators that are crucial to disease progression. The identification of the structural basis of EP300/CBP inhibition by CCS1477 provides deep insights suitable for the derivation of second generation bromodomain inhibitors with enhanced BBB penetration that should be more appropriate for targeting this disease. Our results further provide insights for

ongoing clinical trial and medicinal chemistry efforts targeting subdomains of these proteins.

## Methods

### General
These studies complied with all relevant ethical regulations, and were approved by the St. Jude Children's Research Hospital Institutional Biosafety Committee. Animal studies were approved by the St. Jude Children's Research Hospital Animal Care and Use Committee.

### Cell lines and reagents
Cell lines used in PRISM screening have been previously described[33,34]. 143B cells were obtained from ATCC and cultured in Eagle's MEM with 0.015 mg/mL 5-bromo-2'-deoxyuridine and 10% FBS. NCIH650 and NCIH2122 cells were obtained from ATCC and cultured in RPMI with 10% FBS. Kelly cells were obtained from DSMZ and cultured in RPMI with 10% FBS. RhJT and TE617T cells were a gift of the Broad Institute Pediatric Dependencies Project and cultured in RPMI with 10% FBS. HDMB03 and MB002 cells were generously provided by Till Milde (KiTZ, Heidelberg) and Yoon-Jae Cho (OHSU), respectively, and cultured in neurosphere medium as previously described[49]. All cells were validated to be free of *mycoplasma spp.* with routine testing for identity by short-tandem repeat testing.

### PRISM and bromoscan analysis
Dose-response sequencing data was processed to determine area-under-the-curve (AUC) values, as previously described[33,34]. Tumor type assignments for cell lines were determined using the Cancer Cell Line Encyclopedia annotations[68]. Data was filtered on a per-cell line basis to determine AUC values for both CCS1477 and A485, prior to comparison. Tumor grouping analyses were restricted to those with >3 representative cell lines. AUC data for each compound was normalized across all cell lines treated by calculating the median AUC per treatment and then median normalizing the data. Bromoscan assay was performed by Eurofins DiscoverX (San Francisco, CA), using 1 µM concentrations of CCS1477.

### CellTiter-Glo assay
CellTiter-Glo assay was performed as per the manufacturer's instructions, at the noted timepoint after compound treatment or plating. For Kelly, TE617T, RhJT, NCIH2122, NCIH650 and 143B, 500 cells were plated in normal growth media per well in a 384-well plate (Corning). HDMB03 and MB002 cells were plated in growth media at 500 cells/well in a 384-well plate. Compounds were dispensed in a dose range from 1 nM to 10 µM using a Tecan D300e compound dispenser (Tecan). Cells were incubated for the noted timepoints prior to performing CellTiter-Glo assay as per the manufacturer's instructions, and reading plates on an Envision 2104 microplate reader.

### Western blotting and affinity pulldown assays
Western blotting protocols were as previously described[16,69]. Briefly, cells growing in culture were lysed for whole-cell lysates using RIPA buffer, or for affinity pulldown using IP-lysis buffer (Pierce Biotechnology) per the manufacturer's protocol. Equal amounts of protein was affinity purified using biotinylated CCS1477 with or without other compounds (CCS1477, CCS1477-int(1)) overnight, prior to recovery using streptavidin-coated dynabeads (Thermo Fisher Scientific) as per the manufacturer's instructions. Equal amounts of protein were resolved by Western blotting using 4% to 12% Bis−Tris NuPAGE gels (Thermo Fisher Scientific) prior to transfer and immunoblotting using primary antibodies to EP300 (1:1000, Abcam ab10485), BRD4 (1:1000, Epicypher 13-2003), c-MYC (1:1000, Cell Signaling Technology #5605) or β-Actin (1:1000, Cell Signaling Technology #4967). Secondary antibodies were horseradish peroxidase−conjugated anti-rabbit or anti-mouse (1:5000, Santa Cruz Biotechnology, sc-2357, sc-

358914), incubated prior to exposure to enhanced chemiluminescence reagents (GE Amersham). Densitometry was performed using Image Lab v6.1 (BioRad).

**Protein expression and purification.** Expression plasmids (pNIC28-Bsa4) for the bromodomain of EP300 (residues 1048-1161), CBP (residues 1081-1197), and BRD4-BD1 (residues 44-168) were obtained from Addgene, transformed into E. coli BL21 (DE3) RIL expression cells and grown at 37 °C in LB medium (Fisher Scientific) containing carbenicillin (0.1 mg/mL) and chloramphenicol. At OD600 of 0.6, the culture was cooled down to 18 °C and induced with 0.1 mM IPTG. After 18 h growth, the culture was harvested by centrifugation at $6000 \times g$ for 25 min and stored at -80 °C. Harvested cell pellets were re-suspended in 50 mM Na/K phosphate buffer (pH 7.4) containing 100 mM NaCl, 40 mM imidazole, 0.01% w/v lysozyme and 0.01% v/v Triton X-100 at 4 °C for 1 h, subjected to sonication and the lysate was clarified by centrifugation (30,000 × $g$ for 45 min at 4 °C). Proteins were purified by FPLC at 4 °C using columns and chromatography materials from GE Healthcare. The lysate was subjected to an immobilized $Ni^{2+}$ affinity chromatography column equilibrated with 50 mM Na/K phosphate buffer (pH 7.4) containing 100 mM NaCl and 40 mM imidazole using a gradient from 40 to 500 mM of imidazole. Fractions containing the target protein were combined and incubated overnight with TEV protease at 4 °C, and the cleaved 6xHis-tag was removed by a second $Ni^{2+}$ affinity column. Proteins were purified to homogeneity by size exclusion chromatography using Superdex 75 equilibrated with elution buffer (50 mM HEPES pH 7.5, 100 mM NaCl, 2 mM DTT). All BRDs eluted as monomeric protein and were of crystallization grade quality (>95% purity as judged by SDS-PAGE). Pool fractions were combined, concentrated to 10–12 mg/ml, and aliquots were flash-frozen in liquid $N_2$ and stored at −80 °C.

**Crystallization and X-ray crystallography.** All crystallization experiments were performed at 18 °C. Aliquots of purified EP300, CBP or BRD4-BD1 were set up for crystallization screening using a Mosquito liquid handler (TP Labtech) in 200 nl sitting droplets. For co-crystallization, compound was pre-mixed with protein on ice and then diluted in a 1:1 ratio with precipitant to achieve a final concentration of 1 mM in 10% DMSO. Crystals were cryoprotected using the well solution supplemented with 25% ethylene glycol and flash frozen in liquid nitrogen. X-ray diffraction data were collected at −180 °C in house (CuKα X-rays generated by a Rigaku Micro-Max 007-HF X-ray generator), and at the Synchrotron beamlines 22-ID/BM (SER-CAT) and 23-ID (GM/CA) of the Advanced Photon Source, Argonne National Laboratory. Data were reduced and scaled with XDS versions Jan 31, 2020 (BUILT = 20200131), Feb 5, 2021 (BUILT = 20210205), and Jan 10, 2022 (BUILT = 20220820)[70] or HKL2000 version 718.05[71].The structures were solved by molecular replacement, structure refinement was carried out with PHENIX version 1.20.1_4487[72] and model building with Coot version 0.9.6[73]. Initial models for the small molecule ligands were generated with ligand restraints from eLBOW of the PHENIX suite. Figures were prepared using PyMOL version 2.4.1 (Schrödinger, LLC). The coordinate sets and structure factors were deposited in the Protein Data Bank (PDB) (Supplementary Tables 5 and 6).

**Differential scanning fluorimetry (DSF).** DSF experiments were performed with an Applied Biosystem StepOnePlus real-time PCR system (Thermo Fisher Scientific) using 96-well format plates, assayed in quadruplicate. For thermal shift determination, 5 μM protein was mixed with 100 μM compound in 50 mM HEPES, pH 7.5, 150 mM NaCl, 2 mM DTT, 2% DMSO, 5X SYPRO Orange (Thermo Fisher Scientific) in 20 μL reaction volumes. Reaction mixtures were heated from 25 to 95 °C at 1 °C/min with fluorescence readings every 0.5 °C at 610 nm. The observed thermal shift ($\Delta T_m$) was recorded as the difference between the $T_m$ of sample and DMSO reference wells.

**Isothermal calorimetry (ITC).** All experiments were conducted using an PEAQ-ITC calorimeter (Malvern Scientific). BRDs were buffer exchanged using PD10 columns (GE life sciences) into 50 mM HEPES pH7.5 and 100 mM NaCl (ITC buffer) before the experiment and concentrated to ~10 mg/mL. Experiments were carried out in ITC buffer while stirring at 750 rpm through reverse titration. The microsyringe (40 μL load volume) was loaded with the protein sample (250–600 μM protein in ITC buffer) and inserted into the calorimetric cell (0.2 mL cell volume) consisting of compound (20–60 μM in ITC buffer). All titrations were conducted using an initial control injection of 0.2 μL followed by 13 identical injections (-3 μL per injection) with a duration of 2 s (per injection) and a spacing of 150 s between injections. The ratio of protein/compound was optimized to ensure complete saturation of the titrant before the final injection to ensure proper baseline determination. Data were corrected for dilution and analyzed using the MicroCal Origin software to determine the enthalpies of binding ($\Delta H$) and binding constants as described[74]. Thermodynamic parameters were calculated using the basic equation of thermodynamics ($\Delta G = \Delta H - T\Delta S = -RT\ln K_B$) where $\Delta G$, $\Delta H$ and $\Delta S$ are the changes in free energy, enthalpy and entropy of binding, respectively. A single binding site model was used for all ligand-protein interactions.

**Microscale thermophoresis (MST).** MST experiments were performed in a Monolith NT.115 Pico instrument (NanoTemper Technologies) following published procedures[75]. Measurements were performed at medium MST power and 20% excitation power (auto-detect) at 25 °C with standard capillaries using a constant concentration of GFP-tagged CBP (50 nM) and increasing concentrations of inhibitor using a 16 point 2× serial dilution (1.5 nM to 100 μM) in 50 mM HEPES pH 7.5, 150 mM NaCl, 0.05% Tween and 2% DMSO. MST traces were recorded using standard parameters: 5 s MST power off, 30 s MST power on and 5 s MST power off. Measurements were taken at -1-0 s (cold region) and 10 s (hot region) and data were analyzed using the NTAnalysis software. The macroscopic dissociation constant ($K_D$) was determined using the equation: $[BL]/[B0] = (([L0] + [B0] + K_D) - \sqrt{(([L0] + [B0] + K_D)^2 - 4 \times [L0] \times [B0])})/2 \times [B0]$, where [B0] corresponds to the total concentration of target binding sites, [L0] to the concentration of titrated ligand, and [BL] to the concentration of formed complex between ligand and target binding sites[76].

## Spike-in RNA-sequencing and data analysis

Cells were treated with CCS1477, JQ1, A485 or matched concentrations of DMSO as a vehicle control, at the noted doses for a total of 6 h, prior to cell counting, isolation and lysis in Trizol (Thermo Fisher Scientific). External RNA consortium control RNAs (ERCC, Ambion) were added directly to Trizol, based on cell number prior to RNA extraction. RNA samples were treated with DNAse I (Invitrogen) prior to recovery. Samples were obtained in biological triplicate. Total RNA was subjected to ribosomal RNA depletion prior to library preparation using the Truseq Stranded Total Library Prep Kit (Illumina) and sequencing on a Nextseq-500 (paired end, 100 bp reads).

Sequences were aligned using HISAT2 (version 2.1.0)[77] in paired-end mode with default parameters to build version hg19 of the human genome to which the sequences of the ERCC synthetic spike-in RNAs (http://tools.invitrogen.com/downloads/ERCC92.fa) had been added. Per-gene expression was quantified using htseq-count[78] with parameters "-i gene_id --stranded=reverse -f bam -m," and version 87 of the canonical GRCh37 gene list from RefSeq to which ERCC coordinates were added. TPM-normalized (Transcripts Per Million) expression was then computed for each gene and synthetic ERCC spike-in RNA. Per-gene exon length was calculated across all exons of all isoforms of each gene. The standard TPM-normalization strategy is: normterm = sum of (read count * read length/exon length) across all genes. TPM = read count * read length/exon length * 1e6/normterm. We used a loess regression to normalize the TPM values across all samples of the batch,

using only the spike-in values to fit the loess by the "affy" R package[79] with function loess.normalize. This function allowed us to perform loess regression on a matrix of TPM values and used the ERCC subset of data for fitting the loess. The output from this was a matrix of normalized TPM values controlled by ERCC spike-ins.

ERCC-normalized expression of each gene after 6 h of CCS1477, A485 or JQ1 was compared against its expression in DMSO-treated samples. Per-gene fold changes were calculated and statistical differential expression analysis seeking significance values for each gene was performed using DESeq2 with default parameters on raw counts of treated versus DMSO conditions, or comparing A485 and CCS1477-treated samples, using the ERCC probe read counts as controlGenes[80]. Genes were deemed significantly differentially expressed based on an adjusted $p$-value of <0.05 from DESeq2 analysis. Genes that were significantly differentially expressed in any treatment or cell line were selected for the k-means cluster heatmap in Fig. 4a (DESeq2 adjusted $p < 0.05$, $n = 5092$). We used the elbow method[81] to determine the k value in k-means cluster analysis.

Significantly differentially expressed genes (DESeq2 adjusted $p < 0.05$) in both cell lines were integrated to identify the union of high-confidence up or downregulated genes for each treatment subset. These genes served as input for analysis in METASCAPE[82]. Subsetted gene lists were used for analysis by GSEA using the Gene Ontology Hallmarks collection in MSigDB[83], PANTHER[53] or the String database[54].

### Dependency-STRING database analysis

Dependency analysis was performed using the 22Q4 release of the DepMap dataset, comprising seven medulloblastoma cell lines: UW228, DAOY, ONS76, D458, D425, D283MED, D341MED. Dependency was defined as a Chronos gene effect score of <−0.5, and gene lists were filtered for dependency in >2/7 medulloblastoma cell lines. Interaction networks were determined using the String database[54], using medium confidence settings, network edges = confidence, no interactors, and hiding disconnected nodes (for JQ1, CCS1477) or showing disconnected nodes (A485). PANTHER was used to define gene annotations[53], with unknown annotations assigned using individual assessment using the Genecards.org resource.

### In vivo pharmacokinetic (PK) studies

All animal studies were approved by the St. Jude Children's Research Hospital Animal Care and Use Committee and performed in accordance with best practices outlined by the NIH Office of Laboratory Animal Welfare. The plasma pharmacokinetic (PK) profile of CCS1477 was evaluated in female CD-1 nude mice (Charles River) at approximately 8–12 weeks in age. CCS1477 was dissolved in 10% DMSO/90% (20% hydroxypropyl beta cyclodextrin (HP-β-CD) in sterile water) at 2.5 mg/mL for a 25 mg/kg dose and 5 mg/mL for a 50 mg/kg dose, each using a 10 mL/kg IP injection. Three survival blood samples were obtained from each mouse via retro-orbital plexus using 70 µL glass microhematocrit capillary tubes (Fisher), and a fourth final sample by cardiac puncture immediately following the third survival sample, all using K-EDTA as anticoagulant. Samples were obtained at various times up to 24 h post-dose, immediately processed to plasma, and stored at −80 °C until analysis. At terminal time points of 8, 16, and 24 h, carcasses were perfused with PBS, brains extracted and rinsed. Samples were immediately stored on dry ice and transferred to −80 °C until analysis.

Brain samples were homogenized using a FastPrep-24 system (MP Biomedicals, Santa Ana, CA), and stored at -80 °C until analysis by qualified liquid chromatography–tandem mass spectrometry (LC-MS/MS) assay. Plasma (CD-1 mouse, KEDTA, BioIVT) and brain homogenate (NSG mouse) calibrators and quality controls were spiked with solutions, corrected for salt content and purity as necessary, prepared in methanol. Plasma and brain homogenate samples were precipitated, and extracted supernatant was analyzed on an AB Sciex ExionLC high performance liquid chromatography system via an AB Sciex ExionLC

autosampler. For quantitation, a linear model ($1/X^2$ weighting) fit the plasma and brian homogenate calibrators across the 1 to 500 ng/mL range, with a correlation coefficient (R) of ≥0.9967. The lower limit of quantitation (LLOQ), defined as a peak area signal-to-noise ratio of 5 or greater verses a matrix blank with internal standard, was 1 ng/mL in plasma and 6 ng/mL in brain. Sample dilution integrity was confirmed. No matrix effects, ion enhancement or suppression, were detected in blank CD-1 mouse plasma or NSG brain homogenates. The intra-run precision and accuracy was ≤5.74% CV and 90.6% to 105%, respectively. Resultant plasma and brain homogenate CCS1477 concentrations were summarized by dosing group and nominal timepoint, and subjected to noncompartmental analysis (NCA) with Phoenix WinNonlin 8.1 (Certara USA, Inc., Princeton, NJ).

### Compound Synthesis

Due to isolation of small quantities of new species that were insufficient to perform full characterization, melting points are not presented here.

Synthesis of compound **1**

In a 100 mL round bottom flask fitted with a stir bar, compound **S1** (1.53 g, 1.0 eq.) was dissolved in dry THF (40 mL) and TEA (0.59 mL). *trans-N*-Boc-1,4-cyclohexanediamine (1.66 g, 1.2 eq.) was added and the resulting mixture was heated at 60 °C for 16 h. After cooling to room temperature (r.t.), the reaction mixture was diluted with water (100 mL), then extracted with ethyl acetate (2 × 100 mL). Combined organic layers were concentrated in vacuo to give the crude product, which was purified by flash chromatography (petroleum ether: ethyl acetate = 3:1) to provide the desired **S2** as yellow solid (1.6 g, 57% yield).

$^1$H NMR (400 MHz, DMSO-$d6$) δ 7.99 (d, $J = 2.2$ Hz, 1 H), 7.93 (d, $J = 7.8$ Hz, 1 H), 7.54 (dd, $J = 8.9$, 2.1 Hz, 1 H), 7.24 (d, $J = 9.1$ Hz, 1 H), 6.82 (d, $J = 7.7$ Hz, 1 H), 3.64−3.53 (m, 1 H), 3.32−3.24 (m, 1 H), 2.38 (s, 3H), 2.20 (s, 3 H), 2.08−1.98 (m, 2 H), 1.87−1.78 (m, 2 H), 1.49−1.28 (m, 13 H).

MS (ESI) calculated. For $C_{22}H_{31}N_4O_5$ $[M + 1]^+$ 431.23, Found: 431.48.

In a 100 mL round bottom flask fitted with a stir bar, compound **S2** (1.6 g, 1.0 eq.) was dissolved in THF (75 mL) and water (75 mL). Ammonia solution 7.0 M in methanol (10.87 mL, 20 eq.) and sodium dithionite (6.634 g, 10 eq) were added and the reaction stirred at rt for 6 h. The resulting mixture was diluted with ethyl acetate (200 mL), washed with 1 M NaOH (100 mL) and brine (100 mL), the organic phase was concentrated in vacuo to give the crude product, which was purified by flash chromatography (petroleum ether: ethyl acetate = 2:1) to provide the desired **S3** as white solid (1.5 g, 100% yield).

$^1$H NMR (400 MHz, DMSO) δ 6.77 (d, $J = 7.7$ Hz, 1 H), 6.52 (d, $J = 1.9$ Hz, 1 H), 6.50−6.42 (m, 2 H), 4.62 (brs, 2 H), 4.28 (d, $J = 7.4$ Hz, 1 H), 3.28−3.19 (m, 1 H), 3.17−3.07 (m, 1 H), 2.33 (s, 3 H), 2.16 (s, 3 H), 2.05−1.97 (m, 2 H), 1.85−1.77 (m, 2 H), 1.38 (s, 9 H), 1.32−1.18 (m, 4 H).

MS (ESI) calculated. For $C_{22}H_{33}N_4O_3$ $[M+1]^+$ 401.25, Found: 401.49.

In a 50 mL round bottom flask fitted with a stir bar, compound **S3** (1.3 g, 1.0 eq.), (S)-6-oxopiperidine-2-carboxylic acid (1.4 g, 1.1 eq.), and DIEA (0.7 mL, 1.2 eq.) were dissolved in 10 mL DMF. HATU (1.4 g, 1.1 eq.) was added and stirred at r.t. overnight. The resulting mixture was diluted with water, then extracted with ethyl acetate, dried over $Na_2SO_4$, filtered and concentrated in vacuo to give crude product, which was purified by flash chromatography (petroleum ether: ethyl acetate = 2:1) to provide the desired **S4** as white solid (1.5 g, 88% yield).

**¹H NMR** (400 MHz, DMSO-d6) δ 9.22 (s, 1 H), 7.61 (d, J = 1.8 Hz, 1 H), 7.14 (d, J = 1.8 Hz, 1 H), 7.02 (dd, J = 8.4, 2.0 Hz, 1 H), 6.81−7.72 (m, 2 H), 4.65 (d, J = 7.2 Hz, 1 H), 3.33−3.29 (m, 1 H), 3.28−3.19 (m, 2 H), 2.35 (s, 3 H), 2.21−2.14 (m, 5 H), 2.06−1.95 (m, 3 H), 1.87−1.74 (m, 4 H), 1.71−1.61 (m, 1 H), 1.38 (s, 9 H), 1.34−1.18 (m, 4 H).

MS (ESI) calculated. For $C_{28}H_{40}N_5O_5$ $[M+1]^+$ 526.30, Found: 526.59.

In a 50 mL round bottom flask fitted with a stir bar, crude compound **S4** (1.5 g, 1.0 eq.) was dissolved in acetic acid (50 mL) then heated at 80 °C overnight. The resulting mixture was concentrated then purified by silica gel chromatography (eluent: dichloromethane/methanol) to give 500 mg (36% yield) the product **1** as white powder.

**¹H NMR** (400 MHz, DMSO-d6) δ 8.17 (s, 1 H), 7.87 (d, J = 8.5 Hz, 1 H), 7.74 (d, J = 1.5 Hz, 1 H), 7.62 (d, J = 1.5 Hz, 1 H), 7.15 (dd, J = 8.4, 1.6 Hz, 1 H), 6.86 (d, J = 7.3 Hz, 1 H), 5.12−5.05 (m, 1 H), 4.46−4.34 (m, 1 H), 3.60−3.48 (m, 1 H), 2.39 (s, 3 H), 2.38 - 2.29 (m, 1 H), 2.28 - 2.23 (m, 2 H), 2.22 (s, 3 H), 2.15 − 2.03 (m, 1 H), 2.01−1.69 (m, 7 H), 1.58−1.43 (m, 2 H), 1.41 (s, 9 H).

MS (ESI) calculated. For $C_{28}H_{38}N_5O_4$ $[M+1]^+$ 508.29, Found: 508.39.

**Synthesis of CCS1477-int(1)-biotin and CCS1477-biotin**

In a 4 mL vial fitted with a stir bar, Compound **1** (9.75 mg, 1.0 eq.) was dissolved in DCM (800 μL) followed by dropwise addition of trifluoroacetic acid (TFA, 200 μL). The reaction was stirred at 25 °C for 1 h, and the solvent was removed. The residue was subjected to the next step reaction without further purification.

In a 4 mL vial fitted with a stir bar, the residue from the last step and N-Biotinyl-NH-(PEG)₂-COOH DIPEA (20 atoms) (13.3 mg, 1 eq.) were dissolved in N,N-Dimethylformamide (DMF, 100 μL), followed by addition of triethylamine (TEA, 13 μL, 5 eq.) and HATU (7.3 mg, 1 eq.). Then the resulting mixture was stirred at 25 °C for 2 h. After the reaction was completed, the mixture was directly purified by silica gel chromatography (eluent: dichloromethane/methanol) to give 12.5 mg the product Biotin probe as white powder (Yield: 68%). MS (ESI) calculated. For $C_{48}H_{72}N_9O_9S$ $[M+1]^+$ 950.52, Found: 950.92.

**CCS1477-Biotin** is prepared in a similar way as **CCS1477-int(1)-biotin**. In a 4 mL vial fitted with a stir bar, the free amine **8** and N-Biotinyl-NH-(PEG)₂-COOH DIPEA (20 atoms) (13.3 mg, 1 eq.) were dissolved in N,N-Dimethylformamide (DMF, 100 μL), followed by addition of triethylamine (TEA, 13 μL, 5 eq.) and HATU (7.3 mg, 1 eq.). Then the resulting mixture was stirred at 25 °C for 2 h. After the reaction was completed, the mixture was directly purified by silica gel chromatography (eluent: dichloromethane/methanol) to give 10.2 mg the product Biotin probe as white powder (Yield: 43 %). MS (ESI) calculated. For $C_{54}H_{74}F_2N_9O_9S$ $[M+1]^+$ 1062.52, Found: 1062.82.

**CCS1477** and compound **2** are synthesized by following the procedure reported in patent WO2019202332 A1.

Synthesis of compound **3**

In a 4 mL vial fitted with a stir bar, Compound **2** (8.5 mg, 1.0 eq.), (3-(difluoromethyl)phenyl)boronic acid (5.2 mg, 1.5 eq.), and Cu-TMEDA catalyst (1.4 mg, 0.15 eq.) were dissolved in acetonitrile (100 μL), followed by addition of 1,8-diazabicyclo[5.4.0]undec-7-ene (DBU, 1 μL, 0.1 eq.). The reaction mixture was stirred at 60 °C for 24 h. The mixture was purified directly by silica gel chromatography (eluent: dichloromethane/methanol) to give 7.9 mg the product **3** as white powder (Yield: 71 %).

**¹H NMR** (500 MHz, CDCl₃) δ 7.61 (d, J = 1.3 Hz, 1 H), 7.37 (d, J = 8.6 Hz, 1 H), 7.29 (s, 1 H), 7.26−7.20 (m, 3 H), 7.00 (dd, $J_1$ = 8.5 Hz, $J_2$ = 1.5 Hz, 1 H), 6.45 (t, J = 56.5 Hz, 1 H), 5.25 (t, J = 5.3 Hz, 1 H), 3.99−3.89 (m, 1 H), 3.32 (s, 3 H), 3.24−3.12 (m, 1 H), 2.85−2.74 (m, 1 H), 2.68−2.57

(m, 1 H), 2.36 (s, 3 H), 2.23 (s, 3 H), 2.41–1.98, 1.95–1.83, 1.80–1.69, 1.39–1.05 (m, 12 H);

MS (ESI) calculated. For $C_{31}H_{35}F_2N_4O_3$ $[M+1]^+$ 549.27, Found: 549.44.

Synthesis of compound **4**

In a 4 mL vial fitted with a stir bar, Compound **2** (8.5 mg, 1.0 eq.), (3,5-bis(trifluoromethyl)phenyl)boronic acid (7.7 mg, 1.5 eq.), and Cu-TMEDA catalyst (1.4 mg, 0.15 eq.) were dissolved in acetonitrile (100 µL), followed by addition of 1,8-diazabicyclo[5.4.0]undec-7-ene (DBU, 1 µL, 0.1 eq.). The reaction mixture was stirred at 60 °C for 24 h. The mixture was purified directly by silica gel chromatography (eluent: dichloromethane/methanol) to give 4.0 mg the product **4** as white powder (Yield: 31 %).

[1]H NMR (500 MHz, CDCl₃) δ 7.67 (s, 2 H), 7.59 (d, $J=1.3$ Hz, 1 H), 7.56 (s, 1 H), 7.37 (d, $J=8.2$ Hz, 1 H), 7.00 (dd, $J_1=8.3$ Hz, $J_2=1.6$ Hz, 1 H), 5.28 (t, $J=5.1$ Hz, 1 H), 4.04-3.92 (m, 1 H), 3.33 (s, 3 H), 3.26-3.15 (m, 1 H), 2.88-2.75 (m, 1 H), 2.69–2.58 (m, 1 H), 2.34 (s, 3 H), 2.20 (s, 3 H), 2.41–1.98, 1.95–1.83, 1.80–1.69, 1.39–1.05 (m, 12 H)

MS (ESI) calculated. For $C_{32}H_{33}F_6N_4O_3$ $[M+1]^+$ 635.25, Found: 635.42.

Synthesis of compound **5**

In a 4 mL vial fitted with a stir bar, Compound **2** (8.5 mg, 1.0 eq.), phenylboronic acid (3.7 mg, 1.5 eq.), and Cu-TMEDA catalyst (1.4 mg, 0.15 eq.) were dissolved in acetonitrile (100 µL), followed by addition of 1,8-diazabicyclo[5.4.0]undec-7-ene (DBU, 1 µL, 0.1 eq.). The reaction mixture was stirred at 60 °C for 24 h. The mixture was purified directly by silica gel chromatography (eluent: dichloromethane/methanol) to give 7.7 mg the product **5** as white powder (Yield: 77 %).

[1]H NMR (500 MHz, CDCl₃) δ 7.62 (s, 1 H), 7.36 (d, $J=8.6$ Hz, 1 H), 7.18 (t, $J=7.6$ Hz, 2 H),

7.10 (t, $J=7.5$ Hz, 1 H), 7.06 (d, $J=7.7$ Hz, 2 H), 6.99 (d, $J=8.6$ Hz, 1 H), 5.21 (t, $J=5.2$ Hz, 1 H), 3.95–3.83 (m, 1 H), 3.31 (s, 3 H), 3.23-3.11 (m, 1 H), 2.85–2.75 (m, 1 H), 2.67–2.57 (m, 1 H), 2.36 (s, 3 H), 2.23 (s, 3 H), 2.41–1.98, 1.95–1.83, 1.80–1.69, 1.39–1.05 (m, 12 H);

MS (ESI) calculated. For $C_{30}H_{35}N_4O_3$ $[M+1]^+$ 499.27, Found: 499.44.

Synthesis of compound **6**

In a 4 mL vial fitted with a stir bar, Compound **2** (8.5 mg, 1.0 eq.), (3-(tert-butyl)phenyl)boronic acid (5.3 mg, 1.5 eq.), and Cu-TMEDA catalyst (1.4 mg, 0.15 eq.) were dissolved in acetonitrile (100 µL), followed by addition of 1,8-diazabicyclo[5.4.0]undec-7-ene (DBU, 1 µL, 0.1 eq.). The reaction mixture was stirred at 60 °C for 24 h. The mixture was purified directly by silica gel chromatography (eluent: dichloromethane/methanol) to give 5.0 mg the product **6** as white powder (Yield: 45 %).

[1]H NMR (500 MHz, CDCl₃) δ 7.63 (d, $J=1.3$ Hz, 1 H), 7.32 (d, $J=8.5$ Hz, 1 H), 7.14 (t, $J=7.7$ Hz, 1 H), 7.08 (d, $J=7.9$ Hz, 1 H), 6.96 (dd, $J_1=8.6$ Hz, $J_2=1.6$ Hz, 1 H), 6.91 (d, $J=7.7$ Hz, 1 H), 6.78 (t, $J=1.5$ Hz, 1 H), 5.12 (dd, $J_1=7.0$ Hz, $J_2=5.2$ Hz, 1 H), 3.90-3.75 (m, 1 H), 3.30 (s, 3 H), 3.18-3.07 (m, 1 H), 2.84–2.73 (m, 1 H), 2.68-2.59 (m, 1 H), 2.33 (s, 3 H), 2.20 (s, 3 H), 2.41–1.98, 1.95–1.83, 1.80–1.69, 1.39–1.05 (m, 12 H), 0.86 (s, 9 H);

MS (ESI) calculated. For $C_{34}H_{43}N_4O_3$ $[M+1]^+$ 555.33, Found: 555.48.

Synthesis of compound **7**

In a 4 mL vial fitted with a stir bar, Compound **2** (8.5 mg, 1.0 eq.), (2,3-dihydrobenzo[b][1,4]dioxin-6-yl)boronic acid (5.4 mg, 1.5 eq.), and Cu-TMEDA catalyst (1.4 mg, 0.15 eq.) were dissolved in acetonitrile (100 µL), followed by addition of 1,8-diazabicyclo[5.4.0]undec-7-ene (DBU, 1 µL, 0.1 eq.). The reaction mixture was stirred at 60 °C for 24 h. The mixture was purified directly by silica gel chromatography (eluent: dichloromethane/methanol) to give 6.3 mg the product **7** as white powder (Yield: 56 %).

[1]H NMR (500 MHz, CDCl₃) δ 7.62 (d, $J=0.9$ Hz, 1 H), 7.38 (d, $J=8.6$ Hz, 1 H), 6.99 (dd, $J_1=8.5$ Hz, $J_2=1.5$ Hz, 1 H), 6.66 (d, $J=2.5$ Hz, 1 H), 6.61 (d, $J=8.6$ Hz, 1 H), 6.48 (dd, $J_1=8.7$ Hz, $J_2=2.3$ Hz, 1 H), 5.13 (t, $J=5.1$ Hz, 1 H), 4.08 (s, 4 H), 3.99–3.88 (m, 1 H), 3.33 (s, 3 H), 3.25–3.14 (m, 1 H), 2.84–2.71 (m, 1 H), 2.63–2.54 (m, 1 H), 2.37 (s, 3 H), 2.24 (s, 3 H), 2.41–1.98, 1.95–1.83, 1.80–1.69, 1.39–1.05 (m, 12 H); MS (ESI) calculated. For $C_{32}H_{37}N_4O_3$ $[M+1]^+$ 557.28, Found: 557.42.

## Reporting summary

Further information on research design is available in the Nature Portfolio Reporting Summary linked to this article.

## Data availability

The RNAseq data generated in this study have been deposited in the Gene Expression Omnibus (GEO) database under SuperSeries accession number GSE233609. The x-ray crystallography data generated in this study have been deposited in the Protein Databank (PDB) database under PDB codes 8FV2, 8FVF, 8FVK, 8FVS, 8FXA, 8FXE, 8FXO, 8GA2 Source data are provided as a source data file with this paper.

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

## Acknowledgements

This work was supported by NIH grants K08-CA245251 (A.D.D), R01-CA286444 (A.D.D), P30-CA021765 (to A.D.D., M.F.R, B.J.A), P30-CA076292 (to Moffitt Comprehensive Cancer Center), P01-CA-96832 (M.F.R). This work was supported by the American Lebanese Syrian Associated Charities, and the St. Jude Children's Research Hospital Collaborative Research Consortium on 3D Genomic Architecture of Pediatric Cancer. A.D.D, J.Q. and M.F.R. are supported by the Alex's Lemonade Stand Foundation. A.D.D. was supported by CureSearch for Children's Cancer, the V Foundation for Cancer Research and the Hyundai Hope on Wheels Foundation. A.D.D. and N.A.M.S. were supported by the Rally Foundation for Childhood Cancer Research. We gratefully acknowledge Drs Mark Hatley, Charles W.M. Roberts and Paul Northcott (St. Jude) for providing critical feedback of the manuscript.

## Author contributions

N.A.M.S. Conceptualization, formal analysis, validation, investigation, methodology, writing-original draft, writing-review and editing. M.B. Conceptualization, formal analysis, validation, investigation, methodology, editing. L.H.S. Formal analysis, compound synthesis, validation, investigation, methodology, editing. Y.Z. Data curation, formal analysis, investigation, methodology and editing. A.M. Formal analysis, investigation, methodology and editing. Y.K. Formal analysis, validation, investigation, methodology and editing. S.N. Formal analysis, validation, investigation, methodology, editing. Q.L. Compound synthesis, validation, investigation, methodology, editing. I.M.D. Formal analysis, investigation, editing. S.R. Formal analysis, investigation, editing. V.G. Formal analysis, investigation, editing. M.G.R. Formal analysis, methodology, investigation, editing. M.A.R. Formal analysis, methodology, investigation, editing. T.W. Formal analysis, methodology, validation, investigation, editing. M.K. Formal analysis, methodology, investigation, editing. J.A.R. Formal analysis, methodology, investigation, editing. Y.W. Formal analysis, methodology, investigation, editing. B.B.F. Formal analysis,

methodology, investigation, editing. B.A.O. Formal analysis, methodology, investigation, editing. B.J.A. Formal analysis, data curation, methodology, investigation, writing-original draft, writing-review, editing. M.F.R. Conceptualization, formal analysis, methodology, investigation, writing-original draft, writing-review, editing, resources. E.S. Conceptualization, formal analysis, methodology, investigation, writing-original draft, writing-review, editing, funding acquisition, resources. J.Q. Conceptualization, formal analysis, methodology, investigation, writing-original draft, writing-review, editing, funding acquisition, resources. A.D.D. Conceptualization, formal analysis, methodology, investigation, writing-original draft, writing-review, editing, funding acquisition, resources.

## Competing interests

J.Q. declares other support from Epiphanes and Talus outside the submitted work. B.J.A. is a shareholder of Syros Pharmaceuticals. A.D.D. is a shareholder of Syros Pharmaceuticals and Foghorn Therapeutics. A.D.D., J.Q., Q.L., N.A.M.S. and L.H.S. declare a related patent for small-molecule inhibitors of EP300/CBP and uses thereof. The remaining authors declare no other competing interests.
