## [Peer Review File · Nature Communications]

Group 3 medulloblastoma transcriptional networks collapse under domain-specific EP300/CBP inhibitionREVIEWER COMMENTS

Reviewer #1 (Remarks to the Author):

In the manuscript “Group 3 medulloblastoma transcriptional networks collapse under domain-specific EP300/CBP 2 inhibition” the authors utilize recently developed EP300/CBP domain specific inhibitors – A485 which targets the HAT domain, and CCS1477 which targets the BRD domain to address the general problem of prior work where inhibition of both domains results in unwanted toxicities.

First the authors perform a screen with these compounds across 460 individual cancer cell lines that span many types of cancer. From this screen and additional low-throughput studies they find that while most cancer cell lines display equal sensitivity to A485 (HAT) and CCS1477 (BRD) inhibition, there are some cell lines from specific tumor types that show trends towards specific sensitivity to A485 (HAT) or CCS1477 (BRD). From this data arises a focus on MB, which shows increased sensitivity to CCS1477 (BRD). This sensitivity is further validated in two G3 MB cell lines in vitro. Unfortunately in vivo PK studies show that the CCS1477 (BRD) compound does not enter the brain efficiently, thus making any in vivo studies on sensitivity irrelevant at the moment.

Further mechanistic work is performed on the CCS1477 (BRD) compound to demonstrate important interactions for its function, and molecular studies (transcriptomic) were carried out to compare the impact of CCS1477 (BRD), A485 (HAT) and JQ1 (BET) on transcriptional networks in G3 MB. The authors note that by using a short (6hr) time point they are looking at acute changes caused by drug treatment. They find that CCS1477 (BRD) treated cells display decreased c-MYC protein levels and transcriptional signatures that indicate decreased MYC signaling. Thus, concluding that the primary effect of CCS1477 (BRD) on G3 MB cells is by targeting key gene networks coordinated by MYC.

Main Comments:

From their screen they focus on a specific tumor type – MB – which show a trend as a group of cell lines. But most of the tumor type groups of cell lines show very large variations among individual cell lines in their sensitivity to A485 (HAT) and CCS1477 (BRD). While this

averages out to “no difference” as a whole (suggesting there is no difference in their sensitivity to A485 (HAT) or CCS1477 (BRD) inhibitors) it may be discarding a lot of interesting biology of intra-tumoral heterogeneity in response. Have any of these differences within tumor type cell lines been examined to determine if they are real or just “noise”? Validation using low throughput studies would help determine this.

In vivo brain PK data is mentioned, but I’m not sure if it was included in the main or supplemental figures?

Is there any correlation between drug sensitivity and MYC(N) expression or alterations (amplifications, etc) - especially if MYC driven networks are the main target.

It seems like the decision to focus on MB limits the overall scope of the work. The transcriptional profiling studies identify MYC targets unique to CCS1477 (BRD) in G3 MB cells compared to the other drugs. Is this a generalizable theme for the CCS1477 (BRD) drug in other cancer types? Are the signatures identified in the A485 (HAT) treated G3 MB cells expected to transfer to other cancer types? How would this compare to cell types that are specifically more sensitive to the A485 (HAT) drug? Are different transcriptional networks targeted by A485 (HAT) in cells that are highly sensitive, or are they the same targets found in the G3 MB cells?

Minor comments:

I would move Figure 1E into supplemental since it just describes the cell types used in the screen.

Overall I think the work is very interesting and well presented / written. Yet it only seemed to scratch the surface of the question the authors are addressing about the domain specific effects of A485 (HAT) and CCS1477 (BRD) in cancer biology. Providing a more comprehensive view would help strengthen the manuscript and greatly advance our overall understanding of their function for future therapeutic applications.

Reviewer #2 (Remarks to the Author):

The authors have employed the previously published method, PRISM that allows pooled screening of mixtures of cancer cell lines by uniquely labelling each cell line to investigate the differential effect of catalytic HAT domain and bromodomain (BRD) inhibitor/ligands on various cancers. They used A485 (HAT) or CCS1477 (BRD) and identified a pattern. Group 3 medulloblastoma (G3MB) cells were especially sensitive to EP300/CBP BRD inhibition. They also investigated the global changes in gene expression. The study is very interesting to readers. The experiments are well designed and performed.

- In comparison of the effect of any two compounds in cells, level of cell permeability could be different and significantly affect the differential effect observation despite potentially similar potency in vitro. Would be nice to add a short paragraph on challenges of the screening method and reproducibility of the data in discussion.

- Authors may want to elaborate more in discussion on the enrichment of MYC binding motifs for CCS1477 downregulated genes being less prominent in JQ1-downregulated genesets.

- Would be useful if the differences of JQ1 and CCS1477 in disrupting small networks of proteins / genes essential for MB cell growth could be explained a bit more in detail.

- The experiments for the limited SAR studies with CCS1477 are well designed and performed. However, as the authors stated as well, none of the synthesized compounds was better than CCS1477 and therefore the study did not contribute to the aim / conclusion of this manuscript.

- 305- 306; In stating the usefulness of the SAR studies, the authors may want to soften the language on application of feasible modifications of CCS1477 parent compound to increase selectivity or facilitating BBB penetrance.

- Discussion in parts reads like a summary of the results section. Authors may want to consider revising.

Reviewer #3 (Remarks to the Author):

In this manuscript, Shendy et al. reported their structural, biochemical, and cellular analyses of a EP300/CBP bromodomain inhibitor CCS1477 in blocking proliferation of cancer cells, particularly cells derived from Group 3 medulloblastoma (G3MB), which is a high-risk pediatric brain tumor with a poor prognosis. EP300/CBP are major histone acetyltransferases (HATs) that function as key transcriptional coactivators through their HAT catalytic domains as well as acetyl-lysine binding bromodomains in facilitating gene transcription in chromatin. EP300/CBP have also been implicated in the pathogenesis of numerous human diseases, particularly cancers. In this study, the authors discovered that the G3MB cell lines' proliferation is especially sensitive to chemical inhibition of the EP300/CBP bromodomain by chemical inhibitor CCS1477 much more so than inhibition of the HAT inhibitor A485. The authors further provided detailed structural and biochemical analysis demonstrating how CCS1477 functions as a highly selective EP300/CBP bromodomain inhibitor over other bromodomains, particularly the first bromodomain of BRD4. They further conducted transcriptomics study of CCS1477 vs. A485 and JQ1 (a pan-BET bromodomain inhibitor) in G3MB cell lines, explaining CCS1477's ability and selectivity (to some extent over A485 and JQ1) in inhibition of oncogenic transcriptional activation such as c-MYC. This study provides new understanding of the structure-activity relationship of CCS1477 as a selective EP300/CBP bromodomain inhibitor and its activity in block cancer cell proliferation, particularly G3MB cell lines. However, the study failed to address the key question as to why G3MB cell lines are more sensitive to chemical inhibition of the EP300/CBP bromodomain than to the HAT catalytic domain. Overall, this study does not provide significant advance over the related subject in the literature. Additionally, there are a few specific issues as described below that should be addressed.

1. In all figures, the individual panel labels in the figure legends (A, B, C...) are not consistent with what are in the figures and the text.
2. In Figure 2b, the curves in red are not annotated.
3. The Western blots are generally poor quality and miss MW markers. For instance, in Figure 2g, it is not clear what the signal for BRD4 is for BRD4 long or short isoform.

4. In Figure 4d, GAPDH signals are not even across different samples. As such, the authors cannot really claim that CCS1477 inhibits c-MYC expression more than JQ1.

Response to the Reviewers:

Reviewer#1:

R1: In the manuscript “Group 3 medulloblastoma transcriptional networks collapse under domain-specific EP300/CBP inhibition” the authors utilize recently developed EP300/CBP domain specific inhibitors – A485 which targets the HAT domain, and CCS1477 which targets the BRD domain to address the general problem of prior work where inhibition of both domains results in unwanted toxicities. First the authors perform a screen with these compounds across 460 individual cancer cell lines that span many types of cancer. From this screen and additional low-throughput studies they find that while most cancer cell lines display equal sensitivity to A485 (HAT) and CCS1477 (BRD) inhibition, there are some cell lines from specific tumor types that show trends towards specific sensitivity to A485 (HAT) or CCS1477 (BRD). From this data arises a focus on MB, which shows increased sensitivity to CCS1477 (BRD). This sensitivity is further validated in two G3 MB cell lines in vitro. Unfortunately in vivo PK studies show that the CCS1477 (BRD) compound does not enter the brain efficiently, thus making any in vivo studies on sensitivity irrelevant at the moment. Further mechanistic work is performed on the CCS1477 (BRD) compound to demonstrate important interactions for its function, and molecular studies (transcriptomic) were carried out to compare the impact of CCS1477 (BRD), A485 (HAT) and JQ1 (BET) on transcriptional networks in G3 MB. The authors note that by using a short (6hr) time point they are looking at acute changes caused by drug treatment. They find that CCS1477 (BRD) treated cells display decreased c-MYC protein levels and transcriptional signatures that indicate decreased MYC signaling. Thus, concluding that the primary effect of CCS1477 (BRD) on G3 MB cells is by targeting key gene networks coordinated by MYC.

Main Comments: From their screen they focus on a specific tumor type – MB – which show a trend as a group of cell lines. But most of the tumor type groups of cell lines show very large variations among individual cell lines in their sensitivity to A485 (HAT) and CCS1477 (BRD). While this averages out to “no difference” as a whole (suggesting there is no difference in their sensitivity to A485 (HAT) or CCS1477 (BRD) inhibitors) it may be discarding a lot of interesting biology of intra-tumoral heterogeneity in response. Have any of these differences within tumor type cell lines been examined to determine if they are real or just “noise”? Validation using low throughput studies would help determine this.

Response: We would like to thank Reviewer #1 for their thoughtful and constructive feedback on our manuscript. We are very pleased to see that our manuscript was clearly received, and that the findings were of interest.

We agree that expanding our findings to examine other tumor types is of interest but exhaustive, pan-tumor studies are outside of the main scope of this manuscript. However, we agree that examining differences within tumor type cell lines is important to do. Prior studies using the PRISM platform indicate high reproducibility of results with independent experiments using the same compounds (Corsello et al. *Nature Cancer* 2020), and, more relevantly, between cell line responses to specific agents targeting the same proteins; this latter case is exemplified by the Aurora Kinases (Yu C et al. *Nature Biotechnology* 2016). Indeed, the similarities in cell line responses to compounds targeting the same protein underpins our excitement in our findings that HAT domain and BRD inhibition of EP300 show distinct phenotypes. To address the reviewers question of differences within tumor type cell lines, specifically in non-G3MB tumors, we initially presented validation of screening results by testing specific cell lines from different lineages (143B osteosarcoma, RhJT rhabdomyosarcoma and Kelly neuroblastoma cells), which demonstrated responses that aligned with our screen results. Here, our screens demonstrated that 143B and RhJT cells were more sensitive to CCS1477 than A485, and that Kelly cells were

more sensitive to A485 than CCS1477, demonstrating that distinct tumor types can possess distinct patterns of response to EP300-inhibition. These data are found in the manuscript as Supplementary Figure 1e,f.

To further explore if there are interesting differences within individual lineages that show a range of responses to these two compounds, we have now nominated two lineages (non-small cell lung carcinoma and rhabdomyosarcoma) with wide variation in response. From each of these lineages, we selected representative cell lines that were predicted to be more responsive to one compound or the other: RhJT (rhabdomyosarcoma) and NCIH650 (non-small cell lung carcinoma) were predicted to be more sensitive to CCS1477 than A485, and TE617T (rhabdomyosarcoma) and NCIH2122 (non-small cell lung carcinoma) were predicted to be more sensitive to A485 than CCS1477. We then performed low-throughput testing of these compounds in these cell lines, which validated the predicted responses. These findings suggest that the results of our screen are likely valid and furthermore provide strong evidence for the reviewer's suggested model, i.e., there are certain models within a generally non-preferential tumor type that are more sensitive to HATi or BRDi. These data are now found in **Figure 1** and **Supplementary Figure 1**. For ease of review, the figure and legend to this are reproduced below. Text revisions to the manuscript have been made, and these are also featured below along with relevant line numbers.

Figure 1b-e. Figure legend excerpt:

B-E. TE617T (B), NCIH2122 (C), RhJT (D), and NCIH650 (E) cells were tested for dose-response effects of CCS1477 and A485 after six days by Cell-Titer Glo assay. $n = 3$ independent biological replicates for each dose. Error bars represent S.E.M. Ratio = median normalized AUC ratio.

Supplementary Figure 1e,f. Figure Legend excerpt.

E,F. 143B (E) and Kelly (F) cells were tested for dose-response effects of CCS1477 and A485 after six days by Cell-Titer Glo assay. $n = 3$ independent biological replicates for each dose. Error bars represent S.E.M. Ratio = median normalized AUC ratio.

From the results section:

Lines 130-140: To orthogonally validate these findings, we examined the effects of A485 and CCS1477 on the growth of cell lines from distinct tumor types that displayed a range of responses to these two compounds. Low-throughput testing in TE617T and RhJT rhabdomyosarcoma, NCIH650 and NCIH2122 non-small cell lung carcinoma, 143B osteosarcoma and Kelly neuroblastoma cell lines (**Fig 1b-e, Extended data Fig 1e,f**) demonstrated, as predicted, that NCIH650, RHJT and 143B cells were more sensitive to CCS1477 than A485 (AUC ratio (CCS1477/A485): 0.68 NCIH650, 0.67 143B, 0.76 RHJT) while in contrast, TE617T, NCIH2122 and Kelly cells were more sensitive to A485 than CCS1477 (AUC ratio (CCS1477/A485): 1.13 TE617T, 1.68 NCIH2122, 1.43 Kelly). These data provide support that the results of these screening experiments are reproducible both within individual lineages (non-small cell lung carcinoma and rhabdomyosarcoma) as well as between a variety of lineages.

R1: *In vivo* brain PK data is mentioned, but I'm not sure if it was included in the main or supplemental figures?

Response: We apologize for the confusion. In the supplementary material, we have included the *in vivo* plasma PK data. We tested the brain PK properties of CCS1477 at 8, 16 and 24h after either 25mg/kg or 50mg/kg i.p. injection. At these timepoints, the concentration of CCS1477 in the brain material was less than 6 ng/mL, which is below our assay's lower limit of quantitation. As such, there are no data to insert. We have clarified this in the text, with the relevant sections appended below. Additionally, we updated **Extended Data Figure 2** for increased clarity and with more precise PK parameter estimates, and expanded details about our PK approaches in the Methods section.

Results section:

Lines 200-206: Since medulloblastoma is a primary brain tumor, we explored the blood-brain barrier penetration of CCS1477 in our mice. We sacrificed treated mice at three timepoints after dosing (8, 16, 24h) and perfused the murine vasculature with PBS to eliminate contaminating blood from the brain. Resultant brain tissues were homogenized in PBS and CCS1477 quantitated with a qualified LC-MS/MS method. At these timepoints and doses, we were unable to quantitate CCS1477 in brain tissue with acceptable precision and accuracy (i.e. results were below the lower limit of quantitation of 6 ng/mL) (**data not shown**).

R1: *Is there any correlation between drug sensitivity and MYC(N) expression or alterations (amplifications, etc) - especially if MYC driven networks are the main target.*

Response: We thank reviewer #1 for these very interesting questions. In seeking answers for them, we have conducted several large-scale analyses and further experiments, which we are pleased to share in this revised manuscript.

First, we sought to answer whether, as reviewer #1 asks, there are correlations between drug sensitivity and molecular properties of MYC family genes. *c-MYC* and *MYCN* are the two most dominantly expressed MYC family genes across the Cancer Cell Line Encyclopedia. We performed correlative analyses using the cell line AUC-specific effects of either A485 or CCS1477 across the 460 profiled cell lines in our screen, and integrated these with molecular data from the Cancer Cell Line Encyclopedia. While statistically significant findings were observed (q values less than 0.05), the Pearson correlation coefficients for these findings were extremely modest (between -0.1 and 0.1). In general, these correlations indicated that cell lines with higher expression of *MYC* genes had stronger responses to CCS1477. However, the correlation coefficients were minor, suggesting that if there were specific correlations between any of these

genetic findings and preferential sensitivities, they were slight. Importantly, however, due to the sheer number of measurements, even despite correction for multiple hypothesis testing, we remain concerned that these results reflect statistical significance, with only minor biological significance. Further, these analyses do not rule out the possibility that multiple combined alterations may predict drug sensitivity (i.e. combinations of copy number amplifications, and methylation for example). However, since the correlations with mRNA expression were weak, and the effect of copy number, LOH, methylation are all predicted to terminate in gene expression, we believe that a multivariate analysis is unlikely to yield different results.

Next, we compared quantitative dependency on MYC(N) for each cell line screened with CRISPR and RNAi by DepMap to determine if dependency correlated with domain-specific sensitivity. There was a small (Pearson correlation <0.1) but statistically significant (Q-value <0.05) correlation between MYC dependency and CCS1477-sensitivity. This pan-tumor-type analysis is likely underpowered to detect such a complex relationship, so we subsetted the data to study only extreme responders to A485 or CCS1477. We selected the top 10% of cell lines most selectively sensitive to CCS1477 or A485, but, even among extreme responders, there was not an exceptional dependence on MYC family members or recurrent MYC-related genetic lesions. This is a challenging question to ask, as MYC family members are pan-tumor lethal, and the MYC family member that is expressed is almost always a strong dependency for survival. These observations also suggest that networks in which MYC plays a role could be different between these different cells and that the effects of these compounds are rooted in more than just MYC or MYCN gene expression. Since these analyses are extensive and negative, we are not presenting the results here. However, we point out in our manuscript several statements which reflect these findings:

Introduction section:

Lines 95-98: Finally, we demonstrate that EP300/CBP BRD inhibitors, in contrast with HAT domain inhibitors or BET BRD inhibitors such as JQ1, causes rapid and selective loss of expression of a dense network of genes required to maintain Group 3 medulloblastoma (G3MB) cell growth, including, but not limited to, the medulloblastoma driver oncogene *c-MYC*.

Results section:

Lines 155-161: As predicted by the known differences between chemical inhibition of two proteins (both EP300 and CBP), and genetic loss of a single protein species (*EP300* or *CBP*), there was no association between the individual CRISPR-cas9 knockout of *EP300* or *CBP* and the relative effect of BRD or HAT domain inhibition (**Extended data Fig 1h,i**). These data indicated that the relative susceptibility to EP300/CBP BRD vs HAT domain inhibition, as determined by the ratio of median normalized AUC is not determined by a single driver and may be multifactorial in nature.

Lines 364-367: We performed a Metascape analysis of the genes significantly downregulated by CCS1477, as compared with DMSO, focusing on the oncogenic signatures dataset from MSigDB. Filtering this for significant enrichment ($\log_{10}Q\text{-value}<-3$), we identified an enrichment of MYC-regulated genesets, in addition to several others (**Fig 4e**).

Lines 374-376: These findings suggested a key enhanced, but perhaps not sole role for c-MYC in driving G3MB early responses to EP300/CBP BRD inhibitor treatment, compared with HAT inhibitor treatment.

Lines 390-394: This analysis demonstrated that the majority of genes required for MB cell growth and disrupted by CCS1477 were involved in a highly interconnected and integrated protein-

protein interaction network involved in coordination of mRNA transcription/cell cycle regulation (red) or RNA metabolism (blue) (**Fig 4g**, terms determined by Gene Ontology analysis).

Discussion section:

Lines 462-466: Analysis of mutation, expression, methylation and dependency data from the Cancer Cell Line Encyclopedia⁶¹ failed to identify a correlation between mutational status of EP300/CBP and sensitivity to either HAT or bromodomain inhibition, including strong relationships to *MYC* genes (**Extended Data Fig 1**). These data indicate that *MYC* expression may be only one component of the response to these compounds.

Lines 511-515: Our observations highlight that inhibition of EP300/CBP bromodomains function beyond regulation of *c-MYC* to include dysregulation of gene networks involved in regulation of transcription and RNA splicing including sequence-specific transcription factors, RNA synthesis, transport and processing genes, and cell cycle genes, many of which are previously known to be linked to MB growth^{40,65}.

R1: *It seems like the decision to focus on MB limits the overall scope of the work. The transcriptional profiling studies identify MYC targets unique to CCS1477 (BRD) in G3 MB cells compared to the other drugs. Is this a generalizable theme for the CCS1477 (BRD) drug in other cancer types? Are the signatures identified in the A485 (HAT) treated G3 MB cells expected to transfer to other cancer types? How would this compare to cell types that are specifically more sensitive to the A485 (HAT) drug? Are different transcriptional networks targeted by A485 (HAT) in cells that are highly sensitive, or are they the same targets found in the G3 MB cells?*

We thank reviewer 1 for these interesting questions. To answer them, we have performed further experiments, which we present below. In brief, we do not believe that the patterns of phenotype and molecular response to HAT- vs. BRD-inhibition are completely similar across tumor types and instead, we believe that G3MB represents a specialized setting that we have explored in some detail. As these findings are unexpected, we believe they fit outside the scope of this current manuscript. Should the reviewer feel that these data are appropriate for inclusion in the manuscript as a supplementary figure, we would be pleased to include them.

Is [MYC targets' more profound response to CCS1477] a generalizable theme for the CCS1477 (BRD) drug in other cancer types?

To address this question, we returned to our analysis of MB cell lines. In our initial submission, we demonstrated by GSEA that *MYC*-target networks were more downregulated by CCS1477 treatment in MB cells, compared with A485 treatment. We also observed that *MYC* expression was more downregulated by CCS1477, compared to A485, in HDMBO3 and MB002 cells at the RNA and protein levels (demonstrated below).

To understand if this was a generalizable theme in other cancer types, we used our low-throughput experiments in tumor cell lines from rhabdomyosarcoma and non-small cell lung carcinoma cells (RhJT, TE617T, NCIH2122, NCIH650) to identify the effects of drug-treatment on *MYC* family member and *MYC*-target gene expression. As with our previous experiments in MB cell lines, we treated cells with the day 3 IC₅₀ dose of each compound, or DMSO as a control for 6h, prior to RNA extraction for ERCC spike-in RNAseq in biological triplicate. We then examined the effect of DMSO, A485 or CCS1477 on expression of *MYC* and *MYC*-target genes. For this analysis, we chose to prioritize the highest expressed *MYC* family gene, which was *c-MYC* for TE617T, NCIH650 and NCIH2122, and *MYCN* for RhJT cells. These results demonstrated that,

in G3MB, *MYC* expression is reduced by CCS1477 and, in the case of MB002 cells, also by A485 (Figure to reviewer, A and B, also found in **Extended Data Figure 8C in the manuscript**). However, in both rhabdomyosarcoma and non-small cell lung carcinoma cell lines, the effect was the opposite - the expression of the highest expressed *MYC* gene was more sensitive to A485 than CCS1477. These observations suggest intrinsic differences between G3MB and other tumor types in terms of relative response to CCS1477 and A485. These data are presented below for reviewer consideration.

Reviewer Figure 1. Effects of CCS1477 and A485 on highest MYC gene expression in six cell lines. Medulloblastoma (A, B), rhabdomyosarcoma (C,D) and non-small cell lung carcinoma (E, F) cell lines were treated with the cell line-specific IC_{50} value of CCS1477 or A485, or DMSO as a control, for 6h prior to lysis and ERCC-controlled RNA-seq analysis. Expression of MYC genes (*c-MYC*, *MYCN*, *MYCL*) was quantitated, and the highest expressed gene shown (all *c-MYC* save for C, RhJT, which expresses *MYCN*). Treatment of medulloblastoma cells with CCS1477 causes loss of *MYC* mRNA, with variable effects on A485. Treatment of other cell lines with CCS1477 causes variable effects on *c-MYC*, from no effect (D) to loss. Treatment with A485 causes enhanced loss of *MYC* mRNA expression, compared with CCS1477. * $p < 0.05$, ** $p < 0.01$, *** $p < 0.001$ by two-way ANOVA. G3MB = group 3 medulloblastoma; RMS = rhabdomyosarcoma; NSCLC = non-small cell lung carcinoma.

Are the signatures identified in the A485 (HAT) treated G3 MB cells expected to transfer to other cancer types?

To answer this question, we identified the cohort of downregulated genes in both HDMBO3 and MB002 cells (G3MB cells) with A485 treatment. This yielded a group of 144 genes ($L2FC < -1$,

adjusted p-value<0.05 by DESEQ2 analysis. We then examined the expression of these genes in rhabdomyosarcoma and non-small cell lung carcinoma cell lines, including those more sensitive to A485 (TE617T and NCIH2122) and those more sensitive to CCS1477 (RHJT and NCIH650). For clarity, we have presented these data below for each individual cell line, visualized as volcano plots.

Reviewer Figure 2. A485 downregulated genes are not similarly downregulated in other tumor lineages. The cohort of significantly downregulated genes (FDR q-value<0.05, Log2FC<|1|) were identified in both Medulloblastoma (HDMBO3, MB002) cells (left). This yielded 144 genes, which are significantly downregulated when visualized by volcano plot (left). Examination of these genes in rhabdomyosarcoma or non-small cell lung carcinoma (right) cells revealed mixed and statistically non-significant patterns.

These data confirm that all 144 genes are downregulated in a statistically significant manner by A485 in HDMBO3 and MB002 G3MB cells (Reviewer Figure 2, top-left and bottom-left). However, in other cell lines, most genes are equivalently affected by A485 and CCS1477. In only a minority of cases are the gene downregulation statistically significant. Further, many of these genes are not well expressed at baseline (TPM<1), indicating that their “downregulation” is unreliable. These data indicate that the effects of A485 on gene expression may be dependent on cell line-specific and perhaps lineage-specific gene expression.

As a corollary, we asked whether similar observations could be made for CCS1477-downregulated genes (Reviewer Figure 3, top-left and bottom-left). We performed the same analysis, identifying 414 genes downregulated by CCS1477 in both HDMBO3 and MB002 cells. As demonstrated below, we observed these genes were statistically downregulated in both MB cell lines, with a more mixed pattern in other cell lines. These results were similar to the findings for A485, reiterating that the effects of CCS1477 on gene expression may be more lineage-specific, or related to the baseline expressed genes in the transcriptome.

Reviewer Figure 3. CCS1477 downregulated genes are not similarly downregulated in other tumor lineages. The cohort of significantly downregulated genes (FDR q-value<0.05, Log2FC<|1|) were identified in both Medulloblastoma (HDMBO3, MB002) cells (left). This yielded 414 genes, which are significantly downregulated when visualized by volcano plot (left). Examination of these genes in rhabdomyosarcoma or non-small cell lung carcinoma (right) cells revealed mixed and statistically non-significant patterns.

Are different transcriptional networks targeted by A485 (HAT) in cells that are highly sensitive, or are they the same targets found in the G3 MB cells?

To address this question, we first identified the expression of genes downregulated by A485 in highly sensitive, non-MB cell lines (TE617T and NCIH2122), using a log₂fold change cutoff of -1 and an adjusted p-value of 0.05. This yielded 383 genes downregulated in TE617T cells, and 602 genes downregulated in NCIH2122 cells. Sixty-four (64) genes overlapped between these two cell lines. We compared these 64 genes to the A485-sensitive genes in G3MB cells (n=144 genes). Only 4 genes (1.9%) were in common, indicating that the transcriptional networks targeted in G3MB are tumor type-specific. We then applied these signatures of 64 genes across the A485-regulated transcriptomes in all other cell lines. These data demonstrated little conservation of expression of these 64 genes across other cell lines, after treatment with A485. Moreover, as in prior analyses, many of these genes are not expressed at baseline, making their quantitation not possible (for example, in MB002 cells). These data indicate that the precise effects of A485 on gene expression are likely context-dependent.

Reviewer Figure 4. A485-regulated genes are likely to be lineage-specific. The cohort of significantly downregulated genes (FDR q -value <0.05 , $\text{Log}_2\text{FC}<|1|$) by A485 in sensitive (TE617T, NCIH2122) cells were identified (n=383 TE617T; n=602 NCIH2122; n=64 overlap). These genes were then examined in other cell lines, which demonstrated mixed patterns to expression.

Due to the negative results of these analyses, we have chosen to present these data in this rebuttal letter, but do not feel it rises to the level of evidence sufficient for inclusion in the primary manuscript. To address what is likely a lineage-specific effect, rather than a pan-cancer effect, we have added comments to the manuscript highlighting these concepts which we reproduce below. We hope the reviewer will agree that these findings do not detract from our manuscripts focus on the networks under selective control of EP300/CBP bromodomains in G3MB.

Introduction Section:

Lines 95-98: Finally, we demonstrate that EP300/CBP BRD inhibitors, in contrast with HAT domain inhibitors or BET BRD inhibitors such as JQ1, causes rapid and selective loss of expression of a dense network of genes required to maintain Group 3 medulloblastoma (G3MB) cell growth, including, but not limited to, the medulloblastoma driver oncogene *c-MYC*.

Results Section:

Lines 155-161: As predicted by the known differences between chemical inhibition of two proteins (both EP300 and CBP), and genetic loss of a single protein species (*EP300* or *CBP*), there was no association between the individual CRISPR-cas9 knockout of *EP300* or *CBP* and the relative effect of BRD or HAT domain inhibition (**Extended data Fig 1h,i**). These data indicated that the relative susceptibility to EP300/CBP BRD vs HAT domain inhibition, as determined by the ratio of median normalized AUC is not determined by a single driver and may be multifactorial in nature.

Discussion Section:

Lines 462-471: Analysis of mutation, expression, methylation and dependency data from the Cancer Cell Line Encyclopedia⁶¹ failed to identify a correlation between mutational status of EP300/CBP and sensitivity to either HAT or bromodomain inhibition, including strong relationships to *MYC* genes (**Extended Data Fig 1**). These data indicate that *MYC* expression may be only one component of the response to these compounds. This concept is echoed by our findings that extensive networks of dependencies are lost in G3MB cells after treatment with CCS1477, compared with A485, which include *MYC* but also other dependency genes. Further, these data are supported by studies demonstrating that while CCS1477 inactivates *MYC* in prostatic carcinoma⁵, other tumors such as acute myeloid leukemia and multiple myeloma have distinct oncogenes affected by CCS1477 treatment³⁷.

Lines 481-483: Further large-scale sequencing efforts are required to correlate these observations and determine the drivers of enhanced reliance on individual subdomains in medulloblastoma and other distinct tumor states, which may be variable.

R1: Minor comments: *I would move Figure 1E into supplemental since it just describes the cell types used in the screen.*

Response: The cell lines that make up PRISM screening, and therefore the lineage distribution of the screen, have undergone evolution since the assay began to be used. As a result, the cell lines profiled in this manuscript represent a different cohort of cell lines to those profiled in other manuscripts. Therefore, we feel that the cell lines that make up the screen are highly relevant and would advocate to keep this panel as a primary figure.

R1: *Overall I think the work is very interesting and well presented / written. Yet it only seemed to scratch the surface of the question the authors are addressing about the domain specific effects of A485 (HAT) and CCS1477 (BRD) in cancer biology. Providing a more comprehensive view would help strengthen the manuscript and greatly advance our overall understanding of their function for future therapeutic applications.*

Response: We thank reviewer #1 for their comments and suggestions. We hope that with our experimental and textual revisions as detailed above, we have been able to develop a more comprehensive view of the domain-specific effects of bromodomain and HAT domain inhibition across cancer. We also agree that the conceptual advances in this paper, namely that domain-specific inhibition can have distinct consequences, are provocative and will hopefully be impactful upon other disease settings. Indeed, we expect to pursue this avenue in greater detail in future work.

Reviewer #2:

General: *The authors have employed the previously published method, PRISM that allows pooled screening of mixtures of cancer cell lines by uniquely labelling each cell line to investigate the differential effect of catalytic HAT domain and bromodomain (BRD) inhibitor/ligands on various cancers. They used A485 (HAT) or CCS1477 (BRD) and identified a pattern. Group 3 medulloblastoma (G3MB) cells were especially sensitive to EP300/CBP BRD inhibition. They also investigated the global changes in gene expression. The study is very interesting to readers. The experiments are well designed and performed.*

Response: We thank reviewer #2 for their encouraging comments on our study and their note that the findings are of interest to readers.

R2: *In comparison of the effect of any two compounds in cells, level of cell permeability could be different and significantly affect the differential effect observation despite potentially similar potency in vitro. Would be nice to add a short paragraph on challenges of the screening method and reproducibility of the data in discussion.*

Response: The reviewer's point is a very important one. Prior work using the PRISM method has demonstrated the high reproducibility of repeat screens, and determined that, in general, reproducibility metrics are excellent, especially when considering secondary validation using high-throughput screening or low-throughput testing (Corsello SM et al. *Nature Medicine* 2020). Due to the manner in which it is performed, the PRISM assay, however, has several characteristics which may cloud interpretation of response, including variability in compound levels of cell permeability between different cell lines and the reliance of cell lines on secreted factors for survival. Therefore, we have added a statement in the discussion to this effect, and it is reproduced below:

Lines 442-449: Previous studies using this screening method demonstrated that experimental repeats conducted with the same compounds³³ or similar compounds affecting the same target³⁴ are highly correlated. Further, low throughput confirmation of selected cell lines in our experiments confirmed our screening results (**Fig 1B-E, Supplementary Fig 1E,F**). In comparing two distinct compounds in high-throughput screening, however, variables that are challenging to control include variability in compound-cell permeability between different cell lines, and the reliance of some cell lines, notably, leukemic cell lines, on secreted factors for survival.

R2: *Authors may want to elaborate more in discussion on the enrichment of MYC binding motifs for CCS1477 downregulated genes being less prominent in JQ1-downregulated genesets.*

Response: We agree with the reviewer that this is an intriguing point. We believe that this effect may be related to temporal effects – that is, CCS1477 effects occur first, prior to JQ1 effects. This is reflected especially in the effects of CCS1477, A485 and JQ1 on MYC protein levels by western blotting, where CCS1477 causes loss of MYC protein prior to other inhibitors (**Fig 4e**). Prior reports have demonstrated that prolonged (24h) treatment with JQ1 in medulloblastoma cell lines also results in loss of MYC expression, suggestive of the notion that JQ1 may take longer to target MYC than CCS1477 does, and as a result, we see earlier enrichment of MYC binding motif loss with CCS1477 than with JQ1. To reflect this, we have added a statement in the discussion to this effect, and it is reproduced below:

Lines 496-504: Prior evidence has implicated a key role for BRD4 inhibition in multiple tumor states, including G3MB^{47,48,51,64} among many others, where dominant phenotypic outcomes on

transcription were related to disruption of *MYC* transcription. Of note, these studies typically have used longer treatments (>24h) of JQ1 or related molecules to elicit effects on *MYC* transcription. In our study, we observed a higher statistical enrichment of *MYC* binding motifs in the promoters of genes downregulated by CCS1477, compared with those downregulated by A485 or JQ1 (**Extended data Fig 6d**). In combination with observations that CCS1477-treatment induces downregulation of *MYC* protein by western blotting (**Fig 4d**), our observations suggest that CCS1477 more rapidly and potently causes disruption of *MYC* mRNA levels than do A485 or JQ1.

R2: *Would be useful if the differences of JQ1 and CCS1477 in disrupting small networks of proteins / genes essential for MB cell growth could be explained a bit more in detail.*

Response: We have expanded this section of the discussion which now reads:

Lines 511-527: Our observations highlight that inhibition of EP300/CBP bromodomains function beyond regulation of *c-MYC* to include dysregulation of gene networks involved in regulation of transcription and RNA splicing including sequence-specific transcription factors, RNA synthesis, transport and processing genes, and cell cycle genes, many of which are previously known to be linked to MB growth^{40,65}. These findings were not observed with HAT domain inhibitors, which demonstrated striking differences with BRD inhibition. Despite having similar outcomes in cell growth at day 3, the transcriptional effects of JQ1 and A485 were strikingly different from those of CCS1477. These differences may reflect either the kinetics by which CCS1477 induces transcriptional dysregulation in G3MB cells, as compared with A485 or JQ1, or the direct targets that are associated with bromodomain or HAT domain activity. At this timepoint, treatment with A485 or JQ1 both induce transcriptional loss of *MYC*, though are insufficient to cause loss at the protein level. Therefore, ongoing studies are aimed at understanding if more prolonged treatment with A485 or JQ1 is sufficient to inactivate similar networks of dependency genes involved in transcription, as are achieved by rapid treatment with CCS1477. Despite this, these studies indicate that domain-specific inhibition may be a general property for consideration in the derivation of targeted therapeutics. These results echo the distinct phenotypes elicited by targeting different bromodomains of BET proteins^{64,66,67}.

R2: *The experiments for the limited SAR studies with CCS1477 are well designed and performed. However, as the authors stated as well, none of the synthesized compounds was better than CCS1477 and therefore the study did not contribute to the aim / conclusion of this manuscript.*

Response: We agree that the SAR did not demonstrate improvement of compound Kd. However, we did observe enhanced binding of some compounds to the CBP bromodomain, compared with BRD4 (**Extended data Fig 3e**). As a result, we believe that the SAR has demonstrated specificity, rather than potency, and have chosen to highlight this in the results. We have edited the text to clarify our conclusion below:

Lines 309-318: While none of the analogues were superior to CCS1477 in terms of binding affinity, substitutions were well tolerated if productive interactions with R¹¹⁷³ were maintained. Among the analogues tested, only compound **6** maintained high activity against CBP but was considerably less potent against BRD4, indicating an increase in target selectivity. Correspondingly, while these compounds displayed increased IC₅₀ values in HDMBO3 G3MB cells (**Table 1**), they also demonstrated enhanced selectivity for the BRD of CBP, compared with BRD4 (**Extended Data Fig 3e**), suggesting more on-target activity against EP300/CBP. Combined, these SAR studies suggest that modifications of the CCS1477 parent compound to

increase efficacy for certain applications, such as improved target selectivity or facilitating BBB penetrance, may be feasible.

R2: 305- 306; *In stating the usefulness of the SAR studies, the authors may want to soften the language on application of feasible modifications of CCS1477 parent compound to increase selectivity or facilitating BBB penetrance.*

Response: We agree and have softened the text appropriately. The old and new text are reproduced below, for comparison (highlighting for emphasis):

Old:

Combined, these SAR studies suggest that modifications of the CCS1477 parent compound to increase efficacy for certain *in vivo* applications, such as improved target selectivity or facilitating BBB penetrance, are feasible.

New:

Lines 316-318: Combined, these SAR studies suggest that modifications of the CCS1477 parent compound to increase efficacy for certain applications, such as improved target selectivity or facilitating BBB penetrance, *may be* feasible.

R2: *Discussion in parts reads like a summary of the results section. Authors may want to consider revising.*

Response: We thank the reviewer #2 for their comments, and have streamlined the discussion throughout to avoid a simple reiteration of the results and to put our results in context.

Reviewer #3

General: In this manuscript, Shendy et al. reported their structural, biochemical, and cellular analyses of a EP300/CBP bromodomain inhibitor CCS1477 in blocking proliferation of cancer cells, particularly cells derived from Group 3 medulloblastoma (G3MB), which is a high-risk pediatric brain tumor with a poor prognosis. EP300/CBP are major histone acetyltransferases (HATs) that function as key transcriptional coactivators through their HAT catalytic domains as well as acetyl-lysine binding bromodomains in facilitating gene transcription in chromatin. EP300/CBP have also been implicated in the pathogenesis of numerous human diseases, particularly cancers. In this study, the authors discovered that the G3MB cell lines' proliferation is especially sensitive to chemical inhibition of the EP300/CBP bromodomain by chemical inhibitor CCS1477 much more so than inhibition of the HAT inhibitor A485. The authors further provided detailed structural and biochemical analysis demonstrating how CCS1477 functions as a highly selective EP300/CBP bromodomain inhibitor over other bromodomains, particularly the first bromodomain of BRD4. They further conducted transcriptomics study of CCS1477 vs. A485 and JQ1 (a pan-BET bromodomain inhibitor) in G3MB cell lines, explaining CCS1477's ability and selectivity (to some extent over A485 and JQ1) in inhibition of oncogenic transcriptional activation such as c-MYC. This study provides new understanding of the structure-activity relationship of CCS1477 as a selective EP300/CBP bromodomain inhibitor and its activity in block cancer cell proliferation, particularly G3MB cell lines. However, the study failed to address the key question as to why G3MB cell lines are more sensitive to chemical inhibition of the EP300/CBP bromodomain than to the HAT catalytic domain. Overall, this study does not provide significant advance over the related subject in the literature. Additionally, there are a few specific issues as described below that should be addressed.

Response: We thank Reviewer #3 for their thoughtful consideration of our manuscript and their constructive questions. We believe that our work provides new understanding of an under-investigated aspect of drug targeting, namely that domain-specific inhibition of the same protein can produce distinct cellular and molecular phenotypes. This observation is of great conceptual importance in cancer biology and beyond, since therapeutic development and drug targets are generally selected based not on the location of inhibitor targeting, but rather on the availability of a targetable domain. The results of our study would indicate that, in the setting of proteins with multiple domains suitable for targeting, the choice of which domain to target is of exceptional importance, to avoid falsely concluding that a target is of limited therapeutic potential.

Here, we are excited about both the opportunity to contribute to an advanced understanding of this biology as well as to explore the roles of an example case, i.e., HAT domain vs. BRD inhibition of essential epigenetic regulators. This proof-of-concept is valuable in the context of medulloblastomas, which have dire prognoses. We agree that there are key next questions that our study raises, including the intrinsic state of these cells that engender domain-inhibition-specific responses. Exploration of these questions is a major component of this manuscript, and we believe we have demonstrated, both in the initial submission and in this resubmission, a major mechanism explaining this response. In short, the intrinsic gene expression programs of G3MB cells are known to rely heavily on MYC family gene expression and regulated networks. In the setting of G3MB, MYC family members respond uniquely to our chemical perturbations, which leads to domain-inhibition-specific gene expression responses, including the downregulation of a network of genes on which G3MB is known to depend. These effects are not solely due to MYC, however, and therefore, future work will include deep characterization of the relative importance of each of these factors—MYC family members, direct MYC targets, and other known G3MB dependencies—in driving the cellular response to domain-specific inhibitors. We have attempted to address these concerns in focused explanation to specific issues (below) and by strengthening the text in the discussion.

R3: In all figures, the individual panel labels in the figure legends (A, B, C...) are not consistent with what are in the figures and the text.

Response: We apologize for this and have made corrections in the revised manuscript.

R3: In Figure 2b, the curves in red are not annotated.

Response: We apologize for this oversight and have included a legend to all three curves in the revised manuscript. For ease of review, the revised figure is reproduced below:

R3: The Western blots are generally poor quality and miss MW markers. For instance, in Figure 2g, it is not clear what the signal for BRD4 is for BRD4 long or short isoform.

Response: We have repeated the western blots in the manuscript to improve their quality, and present revised blots in the revision. In these revisions, we have denoted molecular weights, which we also used to determine whether the BRD4 isoform is the long (MW ~200 kDa) or short (MW ~ 120 kDa) isoforms. In this case, the isoform is the **long** isoform. For ease of review, the revised western blots in the primary manuscript are presented below (save for western blots of MYC expression, which are addressed in the reviewers next comment), and raw western blot figures are included in the resubmission, per the journal guidelines.

Figure Legend Excerpt: **2G.** Biotinylated-CCS1477 pull-downs in HDMB03 cell lysates demonstrates pull-down of EP300 and CBP, but not BRD4 at low concentrations, and interaction with EP300, CBP and BRD4 at higher concentrations of compound. Data is representative of n = 3 independent lysates and reactions.

Figure Legend Excerpt: **3I.** Biotinylated-CCS1477 pull-downs in HDMB03 medulloblastoma cell lysates demonstrates pull-down of EP300 but not BRD4 at low concentrations of CCS1477, and interaction with EP300 and BRD4 at higher concentrations of CCS1477. CCS1477-int(1) fails to interact with either EP300 or BRD4 at either concentration. Data is representative of n = 3 independent lysates and reactions.

R3: In Figure 4d, GAPDH signals are not even across different samples. As such, the authors cannot really claim that CCS1477 inhibits c-MYC expression more than JQ1.

Response: We appreciate the reviewer #3's concern about comparison given uneven loading control signals. To address this, we performed additional western blots on newly extracted protein, with careful attention to protein loading, which resulted in a much clearer western blot picture (which we present below). In this experiment, we had cleaner results with using beta actin as a loading control, and therefore, this is presented in the manuscript. Here, we again observed that CCS1477 inhibits expression of c-MYC more than A485 or JQ1 at this timepoint.

For ease of review, the revised western blot is presented here. Raw western blot figures are included in the resubmission, per the journal guidelines.

Figure Legend Excerpt: **4D**. HDMBO3 cells were treated with A485 (400 nM), CCS1477 (39 nM) or JQ1 (100 nM) for 6h followed by lysis for western blotting for c-MYC proteins. β -actin is demonstrated as a loading control.

REVIEWERS' COMMENTS

Reviewer #1 (Remarks to the Author):

The authors have addressed all of my comments in the original review to a satisfactory degree. The edits provided, especially to the introduction and discussion help provide better interpretation and clarity.

Reviewer #2 (Remarks to the Author):

No additional comment